# Understanding and Robustifying Sub-domain Alignment for Domain Adaptation

## Abstract

In unsupervised domain adaptation (UDA), aligning source and target domains improves the predictive performance of learned models on the target domain. A common methodological improvement in alignment methods is to divide the domains and align sub-domains instead. These sub-domain-based algorithms have demonstrated great empirical success but lack theoretical support. In this work, we establish a rigorous theoretical understanding of the advantages of these methods that have the potential to enhance their overall impact on the field. Our theory uncovers that sub-domain-based methods optimize an error bound that is at least as strong as non-sub-domain-based error bounds and is empirically verified to be much stronger. Furthermore, our analysis indicates that when the marginal weights of sub-domains shift between source and target tasks, the performance of these methods may be compromised. We therefore implement an algorithm to robustify sub-domain alignment for domain adaptation under sub-domain shift, offering a valuable adaptation strategy for future sub-domain-based methods. Empirical experiments across various benchmarks validate our theoretical insights, prove the necessity for the proposed adaptation strategy, and demonstrate the algorithm's competitiveness in handling label shift.

## 1 Introduction

Supervised deep learning has achieved unprecedented success in a wide range of real-world applications. However, obtaining labeled data may be costly, labor-intensive, and/or time-consuming in certain applications, particularly in medical and biological domains (Lu et al., 2017; Li et al., 2020). To this end, unsupervised domain adaptation (UDA) transfers knowledge from a labeled source domain to a different but related unlabeled target domain (Farahani et al., 2021). However, efficient UDA is challenging due to the statistical discrepancies between two domains, hereafter referred to as *domain shift* (Wang & Deng, 2018; Sankaranarayanan et al., 2018; Deng et al., 2019). To address this challenge, much of the UDA research has focused on reducing the distributional gap between the source and target domains (Shen et al., 2018; Liu et al., 2016; Isola et al., 2017; Tzeng et al., 2015; 2017; 2020; Ganin & Lempitsky, 2015; Ganin et al., 2016; Peng et al., 2018). Recent methods further partition the data into sub-domains and align the sub-domains instead (Pinheiro, 2018; Long et al., 2018; Deng et al., 2019). One straightforward definition of the sub-domains is the conditional distributions based on the classification label. Other strategies for defining sub-domains include cross-domain adaptive clustering (Li et al., 2021b), classifier-based backprop-induced weighting (Westfechtel et al., 2023), domain consensus clustering (Li et al., 2021a), joint learning of domain-invariant features and classifiers (Shi & Sha, 2012), and the use of deep clustering (Gao et al., 2020). These sub-domain-based algorithms have shown substantial empirical success. However, the benefits of sub-domain alignments have not been rigorously justified.

In this work, we present a theoretical analysis to establish that the sub-domain based methods are in fact optimizing a generalization bound that is at least as strong as (and empirically much stronger than) the full-domain-based objective functions. Our analysis further reveals that when the marginal weights of the sub-domains shift between source and target, the sub-domain based methods can fail. We then present a novel UDA algorithm, *Domain Adaptation via Rebalanced Sub-domain Alignment* (DARSA), that is motivated by our analysis and addresses the case when marginal sub-domain weights shift. DARSA optimizes reweighted classification error and discrepancy between sub-domains of the source and target tasks. The reweighting scheme follows a simple intuition:

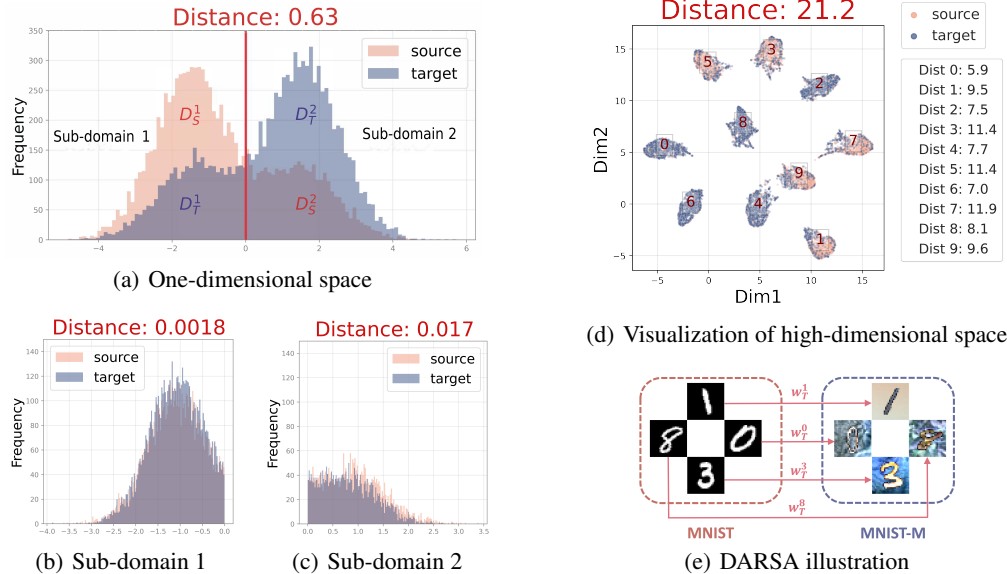

Figure 1: Conceptual overview of our motivation. Listed distances are Wasserstein-1 distances. (a): Representation of data prior to training. The source domain $\mathcal{D}_S$ (coral) consists of two Gaussian centered at $-1.5$ and $1.5$ with weights $0.7$ and $0.3$, respectively. The target domain $\mathcal{D}_T$ (darkblue) is a mixture of two Gaussians centered at $-1.4$ and $1.6$ with inverse weights. We split into subdomains at $x = 0$. (b-c): Representation of data after training. The sub-domain distances are trivial compared to the domain distance in (a). (d) MNIST to MNIST-M UDA task. The features are projected to 2-D with UMAP. The legend indicates distances between corresponding sub-domains (with red sub-domain indices labeled in the figure), and the sub-figure title shows the overall distance. These sub-domain distances are small compared to the overall distance given at the top. (e) DARSA illustration with $w_T^k$ indicating target sub-domain weights, showing DARSA's applicability under label shifting.

*important sub-domains in the target domain need more attention.* To illustrate the concept visually, Figure 1 highlights the strengths of sub-domain alignment, providing insight into how our method operates and the benefits it brings. The contribution of our work is two-fold:

- **Theoretical Contribution:** Our work analyzes and provides a theoretical foundation for sub-domain based methods in domain adaptation, addressing their previous lack of rigorous understanding. Our theoretical framework not only supports our algorithm but can be extended to other methods, contributing to broader impact and value in the field.
- **Algorithmic Contribution:** Our theoretical analysis leads to our algorithm DARSA. DARSA addresses shifted marginal sub-domain weights, which adversely impact existing sub-domain-based methods. We empirically verify its competitive performance under label shifting on various benchmarks, confirming our theoretical insights and validating the proposed adaptation strategy.

## 2 RELATED WORK

We review the most relevant work below and provide a comprehensive discussion in Appendix A.

**Sub-domain-based Domain Adaptation.** Sub-domain alignment has been proven effective in aligning multi-modal distributions and enhancing performance across various tasks (Deng et al., 2019; Long et al., 2018; Pinheiro, 2018; Shi & Sha, 2012; Jiang et al., 2020; Snell et al., 2017). While these methods have demonstrated empirical success, a detailed theoretical perspective on the benefits of incorporating sub-domain structures has yet to be fully explored. Our work complements these existing methodologies by providing a comprehensive theoretical understanding of their inherent advantages. Our theory has the potential to further enhance their overall impact on the field.

**Discrepancy-based Domain Adaptation.** UDA commonly tries to reduce the distribution gap between the source and target domains. One approach to achieve this is discrepancy-based methods in the feature space (Tzeng et al., 2014; Long et al., 2015; Sun et al., 2016), which often use maximum

mean discrepancy (MMD) (Borgwardt et al., 2006). While MMD is a well-known Reproducing Kernel Hilbert Space (RKHS) metric, it is weaker than the Wasserstein-1 distance (Lu & Lu, 2020). Therefore, we use Wasserstein-1 distance in our work.

**Theoretical Analysis of Domain Adaptation.** Many existing domain adaptation methods are inspired by the generalization bounds based on the $\mathcal{H}$-divergence (Ben-David et al., 2006) which is a modified version of the total variation distance that restricts the hypothesis to a given class. These generalization bounds can be estimated by learning a domain classifier with a finite Vapnik–Chervonenkis (VC) dimension. However, this results in a loose bound for most neural networks (Li et al., 2018). In this work, we use the Wasserstein distance for two reasons. First, the Wasserstein-1 distance is upper bounded by the total variation distance (Ben-David et al., 2010), leading to stronger generalization bounds. Additionally, the Wasserstein-1 distance is bounded above by the Kullback-Leibler divergence (a special case of the Rényi divergence when $\alpha$ goes to 1 (Fournier & Guillin, 2015)), giving stronger bounds than those presented by Redko et al (Redko et al., 2017) and Mansour et al (Mansour et al., 2012). Additionally, the Wasserstein distance has stable gradients even when the compared distributions are far apart (Gulrajani et al., 2017).

## 3 PRELIMINARIES

Assume a labeled source dataset $\{(x_S^i, y_S^i)\}_{i=1}^{N_S}$ from a source domain $X_S$ with distribution $P_S$ and an unlabeled target dataset $\{x_T^i\}_{i=1}^{N_T}$ from a target domain $X_T$ with distribution $P_T$. The source dataset has $N_S$ labeled samples, and the target dataset has $N_T$ unlabeled samples. We assume that the samples $x_S^i \in \mathcal{X} \subseteq \mathbb{R}^d$ and $x_T^i \in \mathcal{X} \subseteq \mathbb{R}^d$ are independently drawn from $P_S$ and $P_T$, respectively. The goal is to learn a classifier $f(x)$ that predicts labels $\{y_T^i\}_{i=1}^{N_T}$ for the target dataset. We further assume that $P_S$ and $P_T$ are probability densities of Borel probability measures in the Wasserstein space $\mathcal{P}_1(\mathbb{R}^d)$, i.e., the space of probability measures with finite first moment.

**Sub-domains.** We assume that both $X_S$ and $X_T$ are mixtures of $K$ sub-domains. In other words, we have $P_S = \sum_{k=1}^K w_S^k P_S^k$ and $P_T = \sum_{k=1}^K w_T^k P_T^k$ where we use $P_S^k$ and $P_T^k$ to respectively represent the distribution of the $k$-th sub-domain of the source domain and that of the target domain, and $w_S^k/w_T^k$ correspond to the weights of each sub-domain. Note that $\mathbf{w_S} \doteq [w_S^1, \ldots, w_S^K]$ and $\mathbf{w_T} \doteq [w_T^1, \ldots, w_T^K]$ belong to $\Delta_K$ (the $K - 1$ probability simplex). It is straightforward to define sub-domains as conditional distributions, such that the $k$-th sub-domain is represented as $P_S^k = P(X_S|Y_S = k)$ and $P_T^k = P(X_T|Y_T = k)$, where $Y_S$ and $Y_T$ are the source and target labels, respectively. However, we note that the framework presented in this work is applicable across various sub-domain methods.

**Probabilistic Classifier Discrepancy.** For a distribution $\mathcal{D}$, we define the discrepancy between two functions $f$ and $g$ as:
$$\gamma_{\mathcal{D}}(f, g) = \mathbb{E}_{x \sim \mathcal{D}}\left[|f(x) - g(x)|\right].$$
We use $g_T$ and $g_S$ to represent the true labeling functions of the target and source domains, respectively. We use $\gamma_S(f) \doteq \gamma_{P_S}(f, g_S)$ and $\gamma_T(f) \doteq \gamma_{P_T}(f, g_T)$ to respectively denote the discrepancies of a hypothesis $f$ to the true labeling function for the source and target domains.

**Wasserstein Distance.** The Kantorovich-Rubenstein dual representation of the Wasserstein-1 distance (Villani, 2009) between two distributions $P_S$ and $P_T$ is defined as

$$W_1(P_S, P_T) = \sup_{||f||_L \leq 1} \mathbb{E}_{x \sim P_S}[f(x)] - \mathbb{E}_{x \sim P_T}[f(x)],$$

where the supremum is over the set of 1-Lipschitz functions (all Lipschitz functions $f$ with Lipschitz constant $L \leq 1$. For notational simplicity, we use $D(X_1, X_2)$ to denote a distance between the distributions of any pair of random variables $X_1$ and $X_2$. For instance, $W_1(\Phi(X_S), \Phi(X_T))$ denotes the Wasserstein-1 distance between the distributions of the random variables $\Phi(X_S)$ and $\Phi(X_T)$ for any transformation $\Phi$.

## 4 UNDERSTANDING SUB-DOMAIN-BASED METHODS

We now present our theoretical analysis of sub-domain-based methods, with all proofs deferred to the Appendix B. We first present a generalization bound for domain adaptation that is closely

related to existing work, and then establish a novel generalization bound for sub-domain-based methods, aligning with the objectives used by these existing methods. Furthermore, we demonstrate that the sub-domain-based generalization bound is at least as strong as the non-sub-domain-based generalization bound, which establishes a rigorous theoretical understanding of the advantages of these methods. Our analysis also uncovers that when the marginal weights of sub-domains shift between the source and the target task, sub-domain methods can potentially fail.

## 4.1 GENERALIZATION BOUNDS FOR DOMAIN ADAPTATION

Before presenting our novel theoretical results about sub-domain-based domain adaptation, we first present an upper bound closely related to Ben-David et al. (2010) and Li et al. (2018) Theorem A.8. It is worth noting that we use the Wasserstein-1 distance in our analysis, as it provides a stronger bound than the total variation distance Redko et al. (2017) employed by Ben-David et al. (2010).

**Theorem 4.1** (Full Domain Generalization Bound). *For a hypothesis $f : \mathcal{X} \to [0, 1]$,*

$$\gamma_T(f) \leq \gamma_S(f) + (\lambda + \lambda_H)W_1(P_S, P_T) + \gamma^\star, \tag{1}$$

*where $\gamma^\star = \min_{f \in \mathbb{H}} \gamma_S(f) + \gamma_T(f)$, $\mathbb{H}$ is a hypothesis class included in the set of $\lambda_H$-Lipschitz functions, and the true functions $g_T$ and $g_S$ are both $\lambda$-Lipschitz functions (as defined in Appendix B.1).*

*Remark* 4.2. The upper bound in Theorem 4.1 consists of three components: (i) $\gamma_S(f)$ is the performance of the hypothesis on the source domain, (ii) $W_1(P_S, P_T)$ is the distance between the source and the target domains, and (iii) $\gamma^\star$ is a constant related to the difference between the source and the target problems that cannot be addressed by domain adaptation. For succinctness and clarity of the following analysis, we assume without loss of generality that $\lambda + \lambda_H \leq 1$, simplifying the bound to

$$\gamma_T(f) \leq \gamma_S(f) + W_1(P_S, P_T) + \gamma^\star. \tag{2}$$

Numerous works attempt to solve the domain adaptation problem by designing algorithms that minimize similar generalization bounds to the one in equation 2, e.g., Theorem 1 in Ben-David et al. (2010). These approaches consist of two components: *(i)* a mapping $\Phi : \mathcal{X} \to \mathcal{H}$ that transforms the original problem by embedding $X_S$ and $X_T$ into a shared hidden space $\mathcal{H}$, and *(ii)* a hypothesis $h : \mathcal{H} \to [0, 1]$ for prediction. Since $\gamma_T(h \circ \Phi) = \gamma_{\Phi(X_T)}(h)$, with Theorem 4.1, we have a generalization bound of the function $h \circ \Phi : \mathcal{X} \to [0, 1]$ on the original target problem:

$$\gamma_T(h \circ \Phi) = \gamma_{\Phi(X_T)}(h) \leq \gamma_{\Phi(X_S)}(h) + W_1(\Phi(X_S), \Phi(X_T)) + \gamma_\Phi^\star. \tag{3}$$

If the distance between $\Phi(X_S)$ and $\Phi(X_T)$, i.e., $W_1(\Phi(X_S), \Phi(X_T))$, is close and the classification error of $h$ on the transformed source problem, i.e., $\gamma_{\Phi(X_S)}(h)$, remains low, then the performance of the hypothesis $h \circ \Phi$ on the *original* target problem can be guaranteed. This motivation has led to a variety of domain adaptation frameworks with objectives of the following format:

$$\min_{\substack{\Phi : \mathcal{X} \to \mathcal{H} \\ h : \mathcal{H} \to [0,1]}} \gamma_{\Phi(X_S)}(h) + \alpha \, D(\Phi(X_S), \Phi(X_T)), \tag{4}$$

where $\gamma_{\Phi(X_S)}(h)$ is the classification error of $h$ on the transformed source problem, $D$ is a distance between distributions and $\alpha$ is the balancing weight. In this work, we use Wasserstein-1 distance.

## 4.2 ANALYSIS OF SUB-DOMAIN-BASED METHODS

We first present several results that will be used to build the main theorem. These results themselves may be of interest. First of all, Theorem 4.1 directly leads to the following proposition:

**Proposition 4.3** (Individual Sub-domain Generalization Bound). *For $k \in \{1, \ldots, K\}$, where $K$ represents the total number of distinct sub-domains, for sub-domain $X_S^k$ with distribution $P_S^k$ and $X_T^k$ with distribution $P_T^k$, it holds any $f \in \mathbb{H}$ that*

$$\gamma_T^k(f) \leq \gamma_S^k(f) + W_1(P_S^k, P_T^k) + (\gamma^k)^\star, \tag{5}$$

*where $(\gamma^k)^\star = \min_{f \in \mathbb{H}} \gamma_S^k(f) + \gamma_T^k(f)$, $\mathbb{H}$ is a hypothesis class included in the set of $\lambda_H$-Lipschitz functions, the true functions $g_T$ and $g_S$ are both $\lambda$-Lipschitz functions, and $\lambda + \lambda_H \leq 1$.*

The second result below shows that the classification error of any hypothesis $f$ on a domain can be decomposed into a weighted sum of the classification errors of $f$ on its sub-domains.

**Lemma 4.4** (Decomposition of the Classification Error). *For any hypothesis $f \in \mathbb{H}$,*

$$\gamma_S(f) = \sum_{k=1}^{K} w_S^k \gamma_S^k(f), \gamma_T(f) = \sum_{k=1}^{K} w_T^k \gamma_T^k(f). \tag{6}$$

With above results, we present a generalization bound with sub-domain information:

**Theorem 4.5** (Sub-domain-based Generalization Bound).

$$\gamma_T(f) \leq \sum_{k=1}^{K} w_T^k \gamma_S^k(f) + \sum_{k=1}^{K} w_T^k W_1(P_S^k, P_T^k) + \sum_{k=1}^{K} w_T^k (\gamma^k)^\star. \tag{7}$$

*In particular, in a balanced domain adaptation setting where for all k, $w_S^k = w_T^k$, we have that*

$$\gamma_T(f) \leq \gamma_S(f) + \sum_{k=1}^{K} w_S^k W_1(P_S^k, P_T^k) + \sum_{k=1}^{K} w_S^k (\gamma^k)^\star. \tag{8}$$

*Remark* 4.6. Note that the format of the RHS of equation 8 is reminiscent of the objectives used by the majority of the sub-domain-based methods.

We next show that, under reasonable assumptions, the weighted sum of distances between corresponding sub-domains of the source and target domains is at most as large as the distance between the marginal distribution of the source domain and that of the target domain.

**Theorem 4.7** (Benefits of Sub-domain Alignment). *Under the following assumptions:*

**A1.** *For all k, $P_S^k$ / $P_T^k$ are Gaussian distributions with mean $m_S^k$ / $m_T^k$ and covariance $\Sigma_S^k$ / $\Sigma_T^k$.*
**A2.** *Distance between the paired source-target sub-domain is less or equal to distance between the non-paired source-target sub-domain, i.e., $W_1(P_S^k, P_T^k) \leq W_1(P_S^k, P_T^{k'})$ for $k \neq k'$.*
**A3.** *There exists a small constant $\epsilon > 0$, such that $\max_{1 \leq k \leq K} (tr(\Sigma_S^k)) \leq \epsilon$ and $\max_{1 \leq k \leq K} (tr(\Sigma_T^{k'})) \leq \epsilon$.*

*Then the following inequality holds:*

$$\sum_{k=1}^{K} w_T^k W_1(P_S^k, P_T^k) \leq W_1(P_S, P_T) + \delta_c, \tag{9}$$

*where $\delta_c$ is $4\sqrt{\epsilon}$. In particular, when $w_S^k = w_T^k$ for all k,*

$$\sum_{k=1}^{K} w_S^k W_1(P_S^k, P_T^k) \leq W_1(P_S, P_T) + \delta_c. \tag{10}$$

*Remark* 4.8. In Appendix C, we provide empirical evidence to verify that these assumptions are satisfied on real-world datasets. We note that the assumption of a Gaussian distribution for $X^k$ is not unreasonable since it is often the result of a complex transformation, $\Phi$, and the Central Limit Theorem indicates that the outcome of such a transformation is approximately normally distributed under regularity assumptions (please see Appendix C.1 for empirical evidence).

*Remark* 4.9. $\delta_c$ is a constant dependent only on the variance of the features but not the covariance between features in different dimensions. Moreover, the inequality holds empirically without $\delta_c$ as demonstrated in Figure 3, as well as Figure 7 and Figure 8 in Appendix G.2.

Theorem 4.7 shows that the objective function of sub-domain methods is at least as strong as the objective function of domain alignment methods, explaining its improved performance. However, if the marginal weights of the sub-domain shifts, i.e., $w_S^k \neq w_T^k$, the inequality in equation 10 is not likely to hold and the framework can collapse. One such example is the scenario of shifted label distributions where $w_T^k$ and $w_S^k$ (class weights for target and source domains) can be vastly different. To overcome this, we propose to minimize an objective with the simple intuition that *important sub-domains in the target domain need more attention*. With this motivation, we propose the following objective function for UDA with shifted label distribution:

$$\mathcal{L}(f) = \sum_{k=1}^{K} w_T^k \gamma_S^k(f). \tag{11}$$

In other words, $\mathcal{L}$ reweights the losses of sub-domains so that the sub-domain with more weight in the target domain can be emphasized more. We next prove that through the proposed approach, we can again obtain a sub-domain-based generalization bound that is at least as strong as the full domain generalization bound without the sub-domain information.

**Theorem 4.10.** *Let $\mathbb{H} \doteq \{f | f : \mathcal{X} \to [0,1]\}$ denote a hypothesis space. Under the assumptions in Theorem 4.7, for any $f \in \mathbb{H}$ such that:*

$$\sum_{k=1}^{K} w_T^k \gamma_S^k(f) \leq \sum_{k=1}^{K} w_S^k \gamma_S^k(f), \tag{12}$$

*then we have $\sum_{k=1}^{K} w_T^k (\gamma^k)^\star \leq \gamma^\star$. Further, let*

$$\epsilon_c(f) \doteq \sum_{k=1}^{K} w_T^k \gamma_S^k(f) + \sum_{k=1}^{K} w_T^k W_1(P_S^k, P_T^k) + \sum_{k=1}^{K} w_T^k (\gamma^k)^\star$$

*denote the sub-domain-based generalization bound and let*

$$\epsilon_g(f) \doteq \gamma_S(f) + W_1(P_S, P_T) + \gamma^\star$$

*denote the generalization bound without any sub-domain information, we have,*

$$\epsilon_c(f) \leq \epsilon_g(f) + \delta_c.$$

*Remark* 4.11. In Section 6.1 and Appendix G.2, we provide extensive empirical evidence to establish that equation 12 can easily hold, as the left hand side is the optimization objective. Moreover, in these sections, we offer empirical evidence to further verify the value of this theoretical result by showing that our proposed bound is empirically much stronger than the existing one.

Inspired by our analysis, we propose a framework, *Domain Adaptation with Rebalanced Sub-domain Alignment* (DARSA), for imbalanced UDA, a special case of the sub-domain weight shifting scenario where the class weights of the target domain shifts from that of the source domain.

## 5 METHODS

In DARSA, we divide the source domains into sub-domains based on class labels, and divide target domains into sub-domains using predicted class labels (serving as pseudo labels, which have shown success in previous research (Deng et al., 2019; Lee et al., 2013)) for unlabeled target domains. Motivated by Theorem 4.10, the framework of DARSA, shown in Figure 2, is composed of a source encoder $f_E^S$ parameterized by $\theta_E^S$, a target encoder $f_E^T$ parameterized by $\theta_E^T$, and a classifier $f_Y$ parameterized by $\theta_Y$. The pseudo-code for DARSA can be found in Appendix D.

The objective function of DARSA is defined as follows:

$$\min_{\theta_Y, \theta_E^S, \theta_E^T} \lambda_Y \mathcal{L}_Y + \lambda_D \mathcal{L}_D + \mathcal{L}_C, \tag{13}$$

where $\mathcal{L}_Y$, $\mathcal{L}_D$, $\mathcal{L}_C$ are losses described below with relative weights given by $\lambda_Y$ and $\lambda_D$.

**Weighted source domain classification error $\mathcal{L}_Y$.** The weighted source domain classification error in Theorem 4.10 can be further expressed as:

$$\begin{aligned}
&\sum_{k=1}^{K} w_T^k \gamma_S^k(f) = \sum_{k=1}^{K} w_T^k \int P_S(x|c=k)|f(x) - g_S(x)|dx \\
&= \sum_{k=1}^{K} w_T^k \int \frac{P_S(c=k|x)P_S(x)}{P_S(c=k)}|f(x) - g_S(x)|dx = \sum_{k=1}^{K} \frac{w_T^k}{w_S^k} \mathbb{E}_{x \sim D_s} w_S^k(x)|f(x) - g_S(x)|,
\end{aligned} \tag{14}$$

where variable $c$ represents class, $w_T^k = P_T(c=k)$, $w_S^k = P_S(c=k)$, $w_S^k(x) = P_S(c=k|x)$. We set $P_S(c=k|x) = 1$ only when data point $x$ is in class $k$, otherwise $P_S(c=k|x) = 0$. $w_S^k$ can be set to the marginal source label distribution, and $w_T^k$ can be estimated from the target predictions. From equation 14, $\mathcal{L}_Y(\theta_Y, \theta_E^S)$ is defined as:

$$\mathcal{L}_Y(\theta_Y, \theta_E^S) = \frac{1}{N_S} \sum_{x^i \in \mathcal{X}_S} \mathbb{1}_{y^i=k} \frac{w_T^k}{w_S^k} \ell(\hat{y}^i, y^i),$$

where $\hat{y}^i = f_Y(f_E^S(x^i))$ is the predicted label and $\ell$ can be any non-negative loss function (e.g., cross-entropy loss for classification tasks).

**Weighted source-target subdomain discrepancy $\mathcal{L}_D$.** The weighted source-target domain discrepancy in Theorem 4.10 can be further expressed as:

$$\mathcal{L}_D(\theta_E^S, \theta_E^T, \theta_Y) = \sum_{k=1}^{K} w_T^k W_1(P_S^k, P_T^k) = \sum_{k=1}^{K} w_T^k W_1(f_E^S(x_S^k), f_E^T(x_T^k)), \tag{15}$$

where $x_S^k$ are source samples with labels $y_S = k$, and $x_T^k$ are target samples with predicted labels $\hat{y}_T = k$. We leverage the Sinkhorn algorithm (Cuturi, 2013) to approximate the Wasserstein metric.

**Clustering loss $\mathcal{L}_C$.** The clustering loss $\mathcal{L}_C = \lambda_c \mathcal{L}_{intra} + \lambda_a \mathcal{L}_{inter}$ is comprised of two components: the intra-clustering loss, $\mathcal{L}_{intra}$, and the inter-clustering loss, $\mathcal{L}_{inter}$. The role of $\mathcal{L}_{intra}$ is to satisfy the assumption A.3 in Theorem 4.7. It encourages embeddings of the same label to

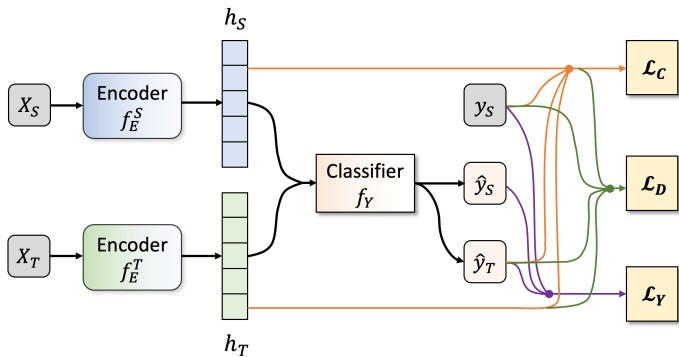

Figure 2: The DARSA framework. Orange lines representing the clustering loss $\mathcal{L}_C$, green lines indicating domain discrepancy $\mathcal{L}_D$, and purple lines indicating source classification loss $\mathcal{L}_Y$.

cluster tightly together, while also pushing embeddings of different labels to separate by at least a user-specified distance, $m$ (Luo et al., 2018). The inter-clustering loss $\mathcal{L}_{inter}$ further enhances sub-domain alignment by aligning the centroids of source sub-domains with those of their corresponding target sub-domains in the representation space. We define $\mathcal{L}_{intra}$ and $\mathcal{L}_{inter}$ as follows:

$$\mathcal{L}_{intra}(\theta_E^S, \theta_E^T, \theta_Y) = \mathcal{L}_{intra}(f_E^S(\mathcal{X}_S)) + \mathcal{L}_{intra}(f_E^T(\mathcal{X}_T)), \qquad (16)$$

$$\mathcal{L}_{intra}(f_E^S(\mathcal{X})) = \frac{1}{N^2} \sum_{i,j=1}^N \left[ \delta_{ij} D_{ij} + (1 - \delta_{ij}) \max\left(0, m - D_{ij}\right) \right];$$

$$\mathcal{L}_{inter}(\theta_E^S, \theta_E^T, \theta_Y) = \frac{1}{K} \sum_{k=1}^K \| \mathcal{C}\big(f_E^S(x_T^k)\big) - \mathcal{C}\big((f_E^T(x_T^k)\big) \|^2, \qquad (17)$$

where $N$ represents the number of samples in the domain $\mathcal{X}$ and $\mathcal{C}(\cdot)$ calculates the centroids of the sub-domains, $\delta_{ij} = 1$ only if $x_i$ and $x_j$ have the same label; otherwise, $\delta_{ij} = 0$. We use the ground truth label or the predicted label if $x$ is in source domain or target domain, respectively. $m$ is a pre-defined distance controlling how separated each sub-domain should be. $D_{ij} = \|f_E(x_i) - f_E(x_j)\|^2$ represents distance between $x_i$ and $x_j$.

## 6 EXPERIMENTS

In this section, we verify our theoretical results and assess DARSA's efficacy through real-world experiments. We begin by empirically confirming the superiority of the sub-domain-based generalization bound (Theorem 4.10) in Section 6.1. Then, we verify that the assumptions for Theorem 4.10 are empirically satisfied on real-world datasets (details in Appendix C). Next, we demonstrate the vital role of subdomain weight re-balancing in Section 6.2 and show DARSA's robustness to minor weight estimation discrepancies. Lastly, given that our theoretical analysis guarantees that DARSA should have competitive performance in scenarios where the number of classes is not overwhelming, we evaluate DARSA on real-world datasets with this property. Comparing with other state-of-the-art UDA baselines, we verify the correctness of our analysis as well as an advantage of DARSA that its strong performance can be guaranteed on particular real-world applications such as those in medical and operations research. We base the following confirmatory experiments on two sets of datasets.

**Experiments on the Digits Datasets.** In our Digits datasets experiments, we evaluate our performance across four datasets: MNIST (M) (LeCun et al., 1998), MNIST-M (MM) (Ganin et al., 2016), USPS (U), and SVHN (S), all modified to induce label distribution shifts. Here, the parameter $\alpha$ denotes the class imbalance rate, representing a ratio such as 1:$\alpha$ and $\alpha$:1 for the odd:even distribution in the source and target datasets, respectively. Weak and strong imbalance correspond to $\alpha = 3$ and $\alpha = 8$. For comprehensive details, refer to Appendix G.

**Experiments on the TST Dataset.** We use the Tail Suspension Test (TST) dataset (Gallagher et al., 2017) of local field potentials (LFPs) from 26 mice with two genetic backgrounds: Clock-$\Delta$19 (a bipolar disorder model) and wildtype. This dataset is publicly available (Carlson et al., 2023). Our

---

[1]The code to replicate all experiments is available at: `https://anonymous.4open.science/r/DARSA/`

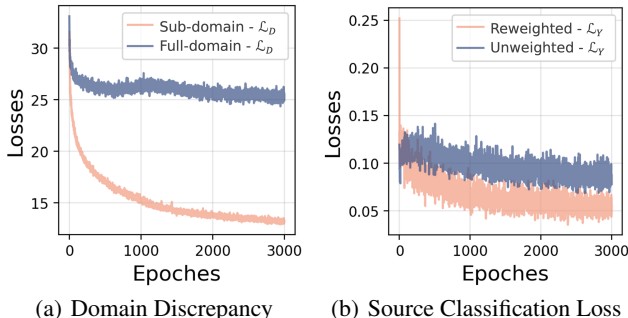

(a) Domain Discrepancy      (b) Source Classification Loss

Figure 3: For MNIST to MNIST-M task under weak imbalance. (a) Compare the domain discrepancy term ($\mathcal{L}_D$) in our proposed bound to that in Theorem 4.1. (b) Compare the source classification term ($\mathcal{L}_Y$) in our proposed bound to that in Theorem 4.1

study involves two domain adaptation tasks, predicting the current condition - home cage (HC), open field (OF), or tail-suspension (TS) - from one genotype to the other. We subsample datasets to induce label distribution shifts with imbalance rate = 2. For comprehensive details, refer to Appendix H.

## 6.1 EMPIRICAL ANALYSIS OF OUR PROPOSED GENERALIZATION BOUND

We first verify the pivotal result in Theorem 4.10 that the sub-domain based generalization bound is at least as tight as the the non-sub-domain bound. We empirically evaluate the proposed bound on the Digits datasets under weak imbalance. As shown in Figure 3, our empirical results demonstrate that the sub-domain-based generalization bound in Theorem 4.5 is empirically much stronger than the non-sub-domain-based bound in Theorem 4.1, corroborating our insights for the effectiveness of sub-domain based methods. Additional experiments on the other UDA tasks in the Digits datasets under weak and strong imbalance also support this claim, and full results are in Appendix G.2.

## 6.2 IMPORTANCE OF RE-WEIGHTING

Here, we experiment on the Digits datasets under weak imbalance to demonstrate the importance of (*i*) weights re-weighting and (*ii*) the accuracy of target sub-domain weights estimation. We compare DARSA with one variation of DARSA which employs uniform weights for all sub-domains and another variation which swaps sub-domain weights estimation of source with target. We also include two other baselines where the weights of the target domain are chosen to be deviating from the truth. Specifically, we compare DARSA with the following configurations:

- DARSA: Full algorithm where weights are inferred.
- DARSA Oracle: Utilizing true values of $w_T^k$.
- DARSA Small Divergence: Setting $w_T^k$ to be 20% divergent from true values.
- DARSA Large Divergence: Setting $w_T^k$ to be 50% divergent from true values.
- DARSA Flip: Swapping $w_T^k$ with $w_S^k$, effectively flipping importance weighting.
- DARSA Uniform: Assigning uniform weights for all sub-domains.

The results of these experiments are in Table 1. We verify the importance of subdomain weights re-balancing by showing that the performance of DARSA degrades significantly without the weights re-balancing or wrong sub-domain weights, further corroborating the value of our insights. Additionally, while the oracle case provides the best performance, inferring the weights in the DARSA algorithm provides nearly the same quality of predictions. In addition, we found our method, DARSA is robust to minor divergence in weights estimation and varying imbalance rates.

## 6.3 DARSA ON REAL-WORLD DATASETS

We now compare DARSA with many competing algorithms on these two datasets. Full details on the experiments, the rationale for competing algorithms choices, and their settings are in Appendix G and Appendix H for the Digits and TST datasets, respectively.

Table 1: Evaluation of the importance of re-weighting on Digits datasets under weak imbalance. Performance is measured by prediction accuracy (%) on the target domain.

|  | $M \to MM$ | $MM \to M$ | $U \to M$ | $S \to M$ |
|---|---|---|---|---|
| DARSA Oracle | **96.2** | 98.4 | **92.7** | **92.6** |
| DARSA Uniform | 67.9 | 96.6 | 75.9 | 71.7 |
| DARSA Small Divergence | 95.6 | 98.3 | 91.4 | 92.4 |
| DARSA Large Divergence | 85.0 | 98.2 | 86.1 | 85.2 |
| DARSA Flip | 55.7 | 65.7 | 57.4 | 65.7 |
| DARSA | 96.0 | **98.8** | 92.6 | 90.1 |

Table 2: Summary of UDA results on the Digits datasets with shifted label distribution, measured in terms of prediction accuracy (%) on the target domain.

|  | $M \to MM$ $\alpha = 3$ | $MM \to M$ $\alpha = 3$ | $U \to M$ $\alpha = 3$ | $S \to M$ $\alpha = 3$ | $M \to MM$ $\alpha = 8$ | $MM \to M$ $\alpha = 8$ | $U \to M$ $\alpha = 8$ | $S \to M$ $\alpha = 8$ |
|---|---|---|---|---|---|---|---|---|
| DANN (Ganin et al., 2016) | 63.1 | 93.0 | 59.8 | 64.9 | 61.1 | 90.2 | 49.1 | 57.3 |
| DSN (Bousmalis et al., 2016) | 62.3 | 98.4 | 59.9 | 15.2 | 57.5 | 95.3 | 30.3 | 17.8 |
| ADDA (Tzeng et al., 2017) | 88.2 | 90.7 | 44.8 | 42.4 | 47.9 | 89.4 | 45.7 | 45.3 |
| pixelDA(Bousmalis et al., 2017) | 95.0 | 96.0 | 72.0 | 68.0 | **81.0** | 95.6 | 29.2 | 60.4 |
| CDAN (Long et al., 2018) | 58.7 | 96.0 | 42.0 | 38.3 | 37.1 | 90.6 | 34.8 | 32.5 |
| WDGRL (Shen et al., 2018) | 60.4 | 93.6 | 63.9 | 64.3 | 22.3 | 91.4 | 46.7 | 52.2 |
| MCD (Saito et al., 2018) | 58.1 | 98.2 | 74.6 | 75.5 | 37.4 | **97.5** | 76.1 | 66.7 |
| CAT (Deng et al., 2019) | 54.1 | 95.4 | 81.0 | 65.8 | 48.9 | 93.8 | 61.3 | 62.2 |
| MDD (Zhang et al., 2019) | 48.7 | 97.7 | 82.3 | 62.4 | 47.6 | 93.6 | 83.2 | 64.5 |
| DRANet (Lee et al., 2021) | 95.2 | 97.8 | 86.5 | 40.2 | 63.3 | 96.1 | 54.2 | 31.3 |
| Source Only | 47.9 | 91.5 | 40.8 | 53.7 | 39.6 | 88.4 | 27.8 | 47.2 |
| DARSA | **96.0** | **98.8** | **92.6** | **90.1** | 78.8 | 97.3 | **87.9** | **83.5** |

**Digits.** Results shown in Table 2 demonstrates DARSA's competitiveness in handling label shifting. Additionally, DARSA performs well with varying imbalance rates (Appendix Table 5) and competes favorably in scenarios without label distribution shifts (Appendix Table 7).

**TST.** As demonstrated in Table 3, DARSA achieves competitive performance on this biologically relevant task. For comprehensive experimental details, refer to Appendix H.

Table 3: Summary of UDA results on the TST datasets with shifted label distribution, measured in terms of prediction accuracy (%) on the target domain.

|  | DANN | WDGRL | DSN | ADDA | CAT | CDAN | Source only | **DARSA** |
|---|---|---|---|---|---|---|---|---|
| Clock-$\Delta$19 to Wildtype | 79.9 | 79.6 | 79.4 | 75.1 | 77.3 | 75.0 | 73.8 | **86.6** |
| Wildtype to Clock-$\Delta$19 | 81.5 | 79.5 | 80.9 | 72.6 | 78.6 | 73.6 | 70.4 | **84.8** |

**Ablation.** To assess the impact of each component within our objective function (Section 5), we conduct an ablation study under weak imbalance. Due to space constraint, the results of this investigation are detailed in Appendix G.6. The ablation analysis confirms that each component in our objective function contributes to the overall performance. Therefore, we recommend the use of all components for optimal results. In addition, we have included feature space visualizations in Appendix E and Appendix Figure 9 which demonstrate that the learned representation of DARSA has improved separation when using all the components, supporting the effectiveness of the proposed objective function.

## 7  CONCLUSION

Sub-domain-based algorithms have demonstrated considerable empirical success across various applications in domain adaptation. However, a comprehensive theoretical understanding of their advantages had been elusive. This work addresses this gap and presents a substantial contribution by providing a rigorous theoretical perspective on the benefits of sub-domain-based methods, thereby potentially enhancing their overall impact in the field. Moreover, our analysis leads to an algorithm DARSA with improved robustness to the shift of sub-domain weights and label distributions.

## REPRODUCIBILITY STATEMENT

Rigorous definitions and complete proofs of our theoretical analysis are included in the Appendix B, with empirical evidence to verify assumptions in Appendix C. The code to replicate all experiments is available at: `https://anonymous.4open.science/r/DARSA/`. Full details on the experiments, competing algorithms, and their settings are in Appendix G and Appendix H for the Digits and TST dataset, respectively. The MNIST, BSDS500, USPS, and SVHN datasets are publicly available with an open-access license. The Tail Suspension Test (TST) dataset (Gallagher et al., 2017) is available to download at `https://research.repository.duke.edu/concern/datasets/zc77sr31x?locale=en` for free under a Creative Commons BY-NC Attribution-NonCommercial 4.0 International license. The experiments are conducted on a computer cluster equipped with a NVIDIA GeForce RTX 2080 Ti that has a memory capacity of 11019MiB.

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

APPENDIX

## A  RELATED WORK

**Discrepancy-based Domain Adaptation.** UDA commonly tries to reduce the distribution gap between the source and target domains. One approach to achieve this is discrepancy-based methods in the extract feature space (Tzeng et al., 2014; Long et al., 2015; Sun et al., 2016), which often use maximum mean discrepancy (MMD) (Borgwardt et al., 2006). While MMD is a well-known Reproducing Kernel Hilbert Space (RKHS) metric, it is weaker than the Wasserstein-1 distance (Lu & Lu, 2020). Therefore, we use Wasserstein-1 distance in our work. Futhermore, many discrepancy-based methods enforce the sharing of the first few layers of the networks between the source and target domains (HassanPour Zonoozi & Seydi, 2022). In contrast, our method allows a more flexible feature space.

**Adversarial-based Domain Adaptation.** Adversarial-based domain adaptation methods aim to encourage domain similarity through adversarial learning (Shen et al., 2018; Liu et al., 2016; Isola et al., 2017; Tzeng et al., 2015; 2017; 2020; Ganin & Lempitsky, 2015; Ganin et al., 2016; Peng et al., 2018; Hoffman et al., 2018). These methods are divided into generative methods, which combine discriminative models with a generating process, and non-generative methods, which use a domain confusion loss to learn domain-invariant discriminative features (Wang & Deng, 2018). However, many existing algorithms fail to align multi-modal distributions under label shifting scenarios. Additionally, training adversarial networks can be challenging due to mode collapse and oscillations (Liang et al., 2018).

**Sub-domain-based Domain Adaptation.** The use of sub-domain adaptation has proven effective in aligning multi-modal distributions, enhancing performance across various tasks (Deng et al., 2019; Long et al., 2018; Pinheiro, 2018; Shi & Sha, 2012; Jiang et al., 2020; Snell et al., 2017). (Deng et al., 2019) introduces the Cluster Alignment with a Teacher (CAT) approach that aligns class-conditional structures across domains. (Long et al., 2018) offers conditional adversarial domain adaptation, enhancing alignment through classifier predictions. (Pinheiro, 2018) proposes an unsupervised domain adaptation approach based on similarity learning, wherein classification is conducted by computing similarities between target domain images and prototype representations of each category. On the other hand, (Shi & Sha, 2012) introduces a method that concurrently learns domain-invariant features and classifiers. (Jiang et al., 2020) elucidates a sampling-based implicit alignment technique, addressing concerns of class imbalance. (Snell et al., 2017) presents prototypical networks designed for few-shot classification, employing distances to class prototype representations for the process. While these methods have demonstrated empirical success, a detailed theoretical perspective on the benefits of incorporating sub-domain structures has yet to be fully explored. Our work aims to complement these existing methodologies by providing a comprehensive theoretical understanding of the advantages inherent in these structures. We believe that such a theoretical perspective, when coupled with the already proven practical success of these methods, holds the potential to further enhance their overall impact on the field.

**Theoretical Analysis of Domain Adaptation.** Many existing domain adaptation methods are inspired by generalization bounds based on the $\mathcal{H}$-divergence (Ben-David et al., 2006). The $\mathcal{H}$-divergence (Ben-David et al., 2006) is a modified version of the total variation distance ($L_1$) that restricts the hypothesis to a given class. These generalization bounds can be estimated by learning a domain classifier with a finite Vapnik–Chervonenkis (VC) dimension. However, this results in a loose bound for most neural networks (Li et al., 2018). In our method, we use the Wasserstein distance for two reasons. First, the Wasserstein-1 distance is bounded above by the total variation distance (Ben-David et al., 2010). Additionally, the Wasserstein-1 distance is bounded above by the Kullback-Leibler divergence (a special case of the Rényi divergence when $\alpha$ goes to 1 (Fournier & Guillin, 2015)), giving stronger bounds than those presented by Redko et al (Redko et al., 2017) and Mansour et al (Mansour et al., 2012). Additionally, the Wasserstein distance has stable gradients even when the compared distributions are far apart (Gulrajani et al., 2017).

**Additional Theoretical Analysis of Domain Adaptation.** Our work contributes to the understanding and improvement of sub-domain alignment methods, a type of popular but yet to be rigorously investigated domain adaptation method. In contrast to our work, (Mansour et al., 2009) studies the adaptation performance of various loss functions and models; (Dhouib et al., 2020) focuses on

the margin violation rate; (Wang et al., 2022) addresses the problem of learning features that align with human understanding of data; (Zhang et al., 2019) proposes generalization theory for classifiers with scoring function and margin loss; (Germain et al., 2016) studies the generalization theory for the weighted majority vote framework; (Blanchard et al., 2021; Albuquerque et al., 2019; Zhao et al., 2018) focus on the setting with multiple source domain.

## B  DEFINITIONS AND PROOFS

**Definition B.1.** For some $K \geq 0$, the set of $K$-Lipschitz functions denotes the set of functions $f$ that verify:
$$\|f(x) - f(x')\| \leq K\|x - x'\|, \ \forall x, x' \in \mathcal{X}$$

In the coming proofs, we assume that the hypothesis class $\mathbb{H}$ is a subset of $\lambda_H$-Lipschitz functions, where $\lambda_H$ is a positive constant, and we assume that the true labeling functions are $\lambda$-Lipschitz for some positive real number $\lambda$.

**Definition B.2.** For a distribution $\mathcal{D}$, we define the discrepancy between two functions $f$ and $g$ as:
$$\gamma_{\mathcal{D}}(f, g) = \mathbb{E}_{x \sim \mathcal{D}}\left[|f(x) - g(x)|\right]$$

We use $g_T$ and $g_S$ to represent the true labeling functions of the target and source domains, respectively. We use $\gamma_S(f) \doteq \gamma_{P_S}(f, g_S)$ and $\gamma_T(f) \doteq \gamma_{P_T}(f, g_T)$ to respectively denote the discrepancies of a hypothesis $f$ to the true labeling function for the source and target domains.

**Definition B.3.** For a distribution $\mathcal{D}$ that can be represented as mixture of $K$ sub-distribution, we define the discrepancy between two functions $f$ and $g$ as:
$$\gamma_{\mathcal{D}}^k(f, g) = \mathbb{E}_{x \sim \mathcal{D}^k}\left[|f(x) - g(x)|\right]$$

where we use $\mathcal{D}^k$ to represent the distribution of the $k$-th sub-distribution.

We use $P_S^k / P_T^k$ to represent the distribution of the $k$-th subdomain of the source domain/target domain respectively. Thus we can use $\gamma_S^k(f) \doteq \gamma_{P_S^k}(f, g_S)$ and $\gamma_T^k(f) \doteq \gamma_{P_T^k}(f, g_T)$ to respectively denote the discrepancies of a hypothesis $f$ to the true labeling function of the $k$-th subdomain of the source domain and that of the target domain.

**Theorem B.4** (Overall Generalization Bound, extending Li et al. (2018) Theorem A.8). *For a hypothesis $f : \mathcal{X} \to [0, 1]$,*
$$\gamma_T(f) \leq \gamma_S(f) + (\lambda + \lambda_H)W_1(P_S, P_T) + \gamma^\star \tag{18}$$
*where $\gamma^\star = \min_{f \in \mathbb{H}} \gamma_S(f) + \gamma_T(f)$, $\mathbb{H}$ is a hypothesis class included in the set of $\lambda_H$-Lipschitz functions, and the true functions $g_T$ and $g_S$ are both $\lambda$-Lipschitz functions (as defined in Definition B.1).*

*Proof.* For a hypothesis $f : \mathcal{X} \to [0, 1]$ with $f \in \mathbb{H}$, we have that
$$\gamma_{P_T}(f, g_T) = \gamma_{P_T}(f, g_T) + \gamma_{P_S}(f, g_S) - \gamma_{P_S}(f, g_S) + \gamma_{P_S}(f, g_T) - \gamma_{P_S}(f, g_T) \tag{19}$$
And then bound the RHS by taking the absolute value of differences:
$$\begin{aligned}
\gamma_{P_T}(f, g_T) &\leq \gamma_{P_S}(f, g_S) + |\gamma_{P_S}(f, g_T) - \gamma_{P_S}(f, g_S)| + |\gamma_{P_T}(f, g_T) - \gamma_{P_S}(f, g_T)| \\
&\leq \gamma_{P_S}(f, g_S) + \mathbb{E}_{x \sim \mathcal{P}_S}[|g_S(x) - g_T(x)|] + |\gamma_{P_T}(f, g_T) - \gamma_{P_S}(f, g_T)|
\end{aligned} \tag{20}$$

As stated in Li et al. (2018), the first two terms proceed exactly as in Ben-David et al. (2010); further derivations are not provided here. We next provide an upper bound for the last term. Let $P_S$ and $P_T$ be the densities of $X_S$ and $X_T$, respectively.

$$|\gamma_{P_T}(f, g_T) - \gamma_{P_S}(f, g_T)| \leq \left| \int (P_T(x) - P_S(x))|f(x) - g_T(x)|dx \right| \tag{21}$$

Since our hypothesis class $\mathbb{H}$ is assumed to be $\lambda_H$-Lipschitz and the true labeling functions are $\lambda$-Lipschitz, we have that for every function $f \in \mathbb{H}$, $h : x \mapsto |f(x) - g_T(x)|$ is $\lambda + \lambda_H$-Lipschitz and it takes its values in $[0, 1]$. Therefore,

$$\begin{aligned}
|\gamma_{P_T}(f, g_T) - \gamma_{P_S}(f, g_T)| &\leq \sup_{h : \mathcal{X} \to [0,1], ||h|| \leq \lambda + \lambda_H} \left| \int (P_T(x) - P_S(x))h(x)dx \right| \\
&= \sup_{h : \mathcal{X} \to [0,1], ||h|| \leq \lambda + \lambda_H} |\mathbb{E}_{x \sim P_T}[h(x)] - \mathbb{E}_{x \sim P_S}[h(x)]|
\end{aligned} \tag{22}$$

Note that due to the symmetric nature of the function space, i.e., if $h$ is K-Lipschitz then $-h$ is K-Lipschitz, we can just pick either side to lead with and drop the absolute value, yielding

$$|\gamma_{P_T}(f, g_T) - \gamma_{P_S}(f, g_T)| \leq (\lambda + \lambda_H)W_1(P_S, P_T) \tag{23}$$

Following the Theorem 2 of Ben-David et al. (2010), we can also easily bound the target error $\gamma_{P_T}(f, g_T)$ by:

$$\gamma_{P_T}(f, g_T) \leq \gamma_{P_S}(f, g_S) + (\lambda + \lambda_H)W_1(P_S, P_T) + \gamma^\star \tag{24}$$

where $\gamma^\star = \min_{f \in \mathbb{H}} \gamma_{P_S}(f, g_S) + \gamma_{P_T}(f, g_T)$.

For succinctness and clarity of the following analysis in this work, as defined in Definition B.2, we express equation 18 as:

$$\gamma_T(f) \leq \gamma_S(f) + (\lambda + \lambda_H)W_1(P_S, P_T) + \gamma^\star \tag{25}$$

where $\gamma^\star = \min_{f \in \mathbb{H}} \gamma_S(f) + \gamma_T(f)$ $\qquad\square$

**Lemma B.5** (Decomposition of the Classification Error). *For any hypothesis $f \in \mathbb{H}$,*

$$\gamma_S(f) = \sum_{k=1}^{K} w_S^k \gamma_S^k(f)$$

$$\gamma_T(f) = \sum_{k=1}^{K} w_T^k \gamma_T^k(f) \tag{26}$$

*Proof.* As stated in Section 3, we assume that both $X_S$ and $X_T$ are mixtures of $K$ sub-domains. In other words, we have $P_S = \sum_{k=1}^{K} w_S^k P_S^k$ and $P_T = \sum_{k=1}^{K} w_T^k P_T^k$ where we use $P_S^k$ and $P_T^k$ to represent the distribution of the $k$-th subdomain of the source domain and that of the target domain respectively and $w_S^k$ and $w_T^k$ correspond to the weights of each sub-domain in the respective domains.

We can write out $\gamma_S(f) = \gamma_{P_S}(f, g_S)$ as sub-domain specific component.

$$\begin{aligned}
\gamma_S(f) &= \gamma_{P_S}(f, g_S) \\
&= \mathbb{E}_{x \sim \mathcal{P}_S}\left[|f(x) - g_S(x)|\right] \\
&= \int P_S(x)|f(x) - g_S(x)|dx \\
&= \int \sum_{k=1}^{K} w_S^k P_S^k |f(x) - g_S(x)|dx \\
&= \sum_{k=1}^{K} w_S^k \int P_S^k |f(x) - g_S(x)|dx \\
&= \sum_{k=1}^{K} w_S^k \mathbb{E}_{x \sim P_S^k}\left[|f(x) - g_S(x)|\right] \\
&= \sum_{k=1}^{K} w_S^k \gamma_{P_S^k}(f, g_S) \\
&\overset{\text{Def.B.3}}{=\joinrel=} \sum_{k=1}^{K} w_S^k \gamma_S^k(f)
\end{aligned} \tag{27}$$

$\qquad\square$

With similar proof, we have:

$$\gamma_T(f) = \gamma_{P_T}(f, g_T) = \sum_{k=1}^{K} w_T^k \gamma_{P_T^k}(f, g_T) = \sum_{k=1}^{K} w_T^k \gamma_T^k(f) \tag{28}$$

**Theorem B.6** (Sub-domain-based Generalization Bound)**.**

$$\gamma_T(f) \leq \sum_{k=1}^{K} w_T^k \gamma_S^k(f) + \sum_{k=1}^{K} w_T^k W_1(P_S^k, P_T^k) + \sum_{k=1}^{K} w_T^k (\gamma^k)^\star \tag{29}$$

*Proof.*

$$\gamma_T(f) \overset{\text{Lemma B.5}}{=} \sum_{k=1}^{K} w_T^k \gamma_T^k(f)$$
$$\overset{\text{Proposition 4.3}}{\leq} \sum_{k=1}^{K} w_T^k \{\gamma_S^k(f, g_S) + W_1(P_S^k, P_T^k) + (\gamma^k)^\star\} \tag{30}$$

$\square$

We next show that, under reasonable assumptions, the weighted sum of distances between corresponding sub-domains of the source and target domains is at most as large as the distance between the marginal distribution of the source domain and that of the target domain.

First we define a Wasserstein-like distance between Gaussian Mixture Models in Definition B.7, which uses Wasserstein-1 distance that extends the Proposition 4 of Delon & Desolneux (2020).

**Definition B.7.** (Wasserstein-like distance between Gaussian Mixture Models) Assume that both $X_S$ and $X_T$ are mixtures of $K$ sub-domains. In other words, we have $P_S = \sum_{k=1}^{K} w_S^k P_S^k$ and $P_T = \sum_{k=1}^{K} w_T^k P_T^k$ where we use $P_S^k$ and $P_T^k$ to represent the distribution of the $k$-th subdomain of the source domain and that of the target domain respectively and $w_S^k$ and $w_T^k$ correspond to the weights of each sub-domain in the respective domains. We define:

$$MW_1(P_S, P_T) = \min_{w \in \Pi(\mathbf{w_S}, \mathbf{w_T})} \sum_{k=1}^{K} \sum_{k'=1}^{K} w_{k,k'} W_1(P_S^k, P_T^{k'}) \tag{31}$$

where $\mathbf{w_S} \doteq [w_S^1, \dots, w_S^K]$ and $\mathbf{w_T} \doteq [w_T^1, \dots, w_T^K]$ belong to $\Delta^K$ (the $K-1$ probability simplex). $\Pi(w_S, w_T)$ represents the simplex $\Delta^{K \times K}$ with marginals $\mathbf{w_S}$ and $\mathbf{w_T}$.

To demonstrate that $MW_1$ is less or equal to the sum of $W_1$ and a term that is dependent on the trace of the covariance matrices of two Gaussian mixtures (extend the proof of Delon & Desolneux (2020)), we start with a lemma. This lemma makes more explicit the distance $MW_1$ between a Gaussian mixture and a mixture of Dirac distributions.

**Lemma B.8** (Extension to Lemma 4.1 of Delon & Desolneux (2020))**.** *Let* $\mu_0 = \sum_{k=1}^{K_0} \pi_0^k \mu_0^k$ *with* $\mu_0^k = \mathcal{N}(m_0^k, \Sigma_0^k)$ *and* $\mu_1 = \sum_{k=1}^{K_1} \pi_1^k \delta_{m_1^k}$. *Let* $\tilde{\mu}_0 = \sum_{k=1}^{K_0} \pi_0^k \delta_{m_0^k}$ *($\tilde{\mu}_0$ only retains the means of* $\mu_0$)*. Then,*

$$MW_1(\mu_0, \mu_1) \leq W_1(\tilde{\mu}_0, \mu_1) + \sum_{k=1}^{K_0} \pi_0^k \sqrt{tr\left(\Sigma_0^k\right)}$$

*where* $\pi_\mathbf{0} \doteq [\pi_0^1, \dots, \pi_0^k]$ *and* $\pi_\mathbf{1} \doteq [\pi_1^1, \dots, \pi_1^k]$ *belong to* $\Delta^K$ *(the $K-1$ probability simplex)*

*Proof.*

$$
\begin{aligned}
MW_1(\mu_0, \mu_1) &= \inf_{w \in \Pi(\pi_0, \pi_1)} \sum_{k,l} w_{k,l} W_1(\mu_0^k, \delta_{m_1^l}) \\
&\leq \inf_{w \in \Pi(\pi_0, \pi_1)} \sum_{k,l} w_{k,l} W_2(\mu_0^k, \delta_{m_1^l}) \\
&= \inf_{w \in \Pi(\pi_0, \pi_1)} \sum_{k,l} w_{k,l} \left[ \sqrt{||m_1^l - m_0^k||^2 + \text{tr}(\Sigma_0^k)} \right] \\
&\leq \inf_{w \in \Pi(\pi_0, \pi_1)} \sum_{k,l} w_{k,l} ||m_1^l - m_0^k|| + \sum_k \pi_0^k \sqrt{\text{tr}(\Sigma_0^k)} \\
&\leq W_1(\tilde{\mu_0}, \mu_1) + \sum_{k=1}^{K_0} \pi_0^k \sqrt{\text{tr}(\Sigma_0^k)}
\end{aligned}
\tag{32}
$$

$\square$

*Remark* B.9. We use $\mu_0$, $\mu_1$, and $\tilde{\mu_0}$ to represent a general scenario for measuring the distance between a Gaussian mixture and a mixture of Diract distributions. In the following proofs, we will utilize the defined notation. For instance, $\mu_0$ can be denoted as $P_S$, while $\tilde{\mu_0}$ corresponds to $\tilde{P}_S$.

**Theorem B.10** (Extension to Proposition 6 in (Delon & Desolneux, 2020)). *Let $P_S$ and $P_T$ be two Gaussian mixtures with $P_S = \sum_{k=1}^K w_S^k P_S^k$ and $P_T = \sum_{k=1}^K w_T^k P_T^k$. For all k, $P_S^k$ / $P_T^k$ are Gaussian distributions with mean $m_S^k$ / $m_T^k$ and covariance $\Sigma_S^k$ / $\Sigma_T^k$. If for $\forall$ k, k', we assume there exists a small constant $\epsilon > 0$, such that $\max_k(trace(\Sigma_S^k)) \leq \epsilon$ and $\max_{k'}(trace(\Sigma_T^{k'})) \leq \epsilon$. then:*

$$
MW_1(P_S, P_T) \leq W_1(P_S, P_T) + 4\sqrt{\epsilon}
\tag{33}
$$

*Proof.* Here, we follow the same structure of the proof for Wassertein-2 in Delon & Desolneux (2020). Let $(P_S^n)_n$ and $(P_T^n)_n$ be two sequences of mixtures of Dirac masses respectively converging to $P_S$ and $P_T$ in $\mathcal{P}_1(\mathbb{R}^d)$. Since $MW_1$ is a distance,

$$
\begin{aligned}
MW_1(P_S, P_T) &\leq MW_1(P_S^n, P_T^n) + MW_1(P_S, P_S^n) + MW_1(P_T, P_T^n) \\
&= W_1(P_S^n, P_T^n) + MW_1(P_S, P_S^n) + MW_1(P_T, P_T^n)
\end{aligned}
$$

We can study the limits of these three terms when n $\to +\infty$

First, observe that $MW_1(P_S^n, P_T^n) = W_1(P_S^n, P_T^n) \underset{n \to +\infty}{\to} W_1(P_S, P_T)$ since $W_1$ is continuous on $\mathcal{P}_1(\mathbb{R}^d)$.

Second, based on Lemma B.8, we have that

$$
MW_1(P_S, P_S^n) \leq W_1(\tilde{P}_S, P_S^n) + \sum_{k=1}^K w_S^k \sqrt{\text{tr}(\Sigma_S^k)} \underset{n \to +\infty}{\to} W_1(\tilde{P}_S, P_S) + \sum_{k=1}^K w_S^k \sqrt{\text{tr}(\Sigma_S^k)}
$$

We observe that $x \mapsto \sqrt{x}$ is a concave function, thus by Jensen's inequality, we have that

$$
\sum_{k=1}^K w_S^k \sqrt{\text{tr}(\Sigma_S^k)} \leq \sqrt{\sum_{k=1}^K w_S^k \text{tr}(\Sigma_S^k)}
$$

Also By Jensen's inequality, we have that,

$$
W_1(\tilde{P}_S, P_S) \leq W_2(\tilde{P}_S, P_S).
$$

And from Proposition 6 in (Delon & Desolneux, 2020), we have

$$
W_2(\tilde{P}_S, P_S) \leq \sqrt{\sum_{k=1}^K w_S^k \text{tr}(\Sigma_S^k)}
$$

Similarly for $MW_1(P_T, P_T^n)$ the same argument holds. Therefore we have,

$$\lim_{n \to \infty} MW_1(P_S, P_S^n) \leq 2\sqrt{\sum_{k=1}^{K} w_S^k \operatorname{tr}(\Sigma_S^k)}$$

And

$$\lim_{n \to \infty} MW_1(P_T, P_T^n) \leq 2\sqrt{\sum_{k=1}^{K} w_T^k \operatorname{tr}(\Sigma_T^k)}$$

We can conclude that:

$$MW_1(P_S, P_T) \leq \lim_{n \to \infty} \inf (W_1(P_S^n, P_T^n) + MW_1(P_S, P_S^n) + MW_1(P_T, P_T^n))$$

$$\leq W_1(P_S, P_T) + 2\sqrt{\sum_{k=1}^{K} w_S^k \operatorname{tr}(\Sigma_S^k)} + 2\sqrt{\sum_{k=1}^{K} w_T^k \operatorname{tr}(\Sigma_T^k)}$$

$$\leq W_1(P_S, P_T) + 4\sqrt{\epsilon}$$

This concludes the proof. □

**Theorem B.11** (Sub-domain Alignment Can Improve Domain Alignment). *If the following assumptions hold:*

**A1.** *For all k, $P_S^k$ / $P_T^k$ are Gaussian distributions with mean $m_S^k$ / $m_T^k$ and covariance $\Sigma_S^k$ / $\Sigma_T^k$.*
**A2.** *Distance between the paired source-target sub-domain is less or equal to distance between the non-paired source-target sub-domain, i.e., $W_1(P_S^k, P_T^k) \leq W_1(P_S^k, P_T^{k'})$ for $k \neq k'$.*
**A3.** *There exists a small constant $\epsilon > 0$, such that $\max_{1 \leq k \leq K}(tr(\Sigma_S^k)) \leq \epsilon$ and $\max_{1 \leq k \leq K}(tr(\Sigma_T^{k'})) \leq \epsilon$.*

*Then the following inequality holds:*

$$\sum_{k=1}^{K} w_T^k W_1(P_S^k, P_T^k) \leq W_1(P_S, P_T) + \delta_c$$

*where $\delta_c$ is $4\sqrt{\epsilon}$.*

*Proof.* With $w \in \Pi(\mathbf{w_S}, \mathbf{w_T})$, we can write out $w_T^k$ as $\sum_{k'=1}^{K} w_{k,k'}$, then based on assumption A.2, we have:

$$\sum_{k=1}^{K} w_T^k W_1(P_S^k, P_T^k) = \sum_{k=1}^{K} \sum_{k'=1}^{K} w_{k,k'} W_1(P_S^k, P_T^k)$$

$$\leq \sum_{k=1}^{K} \sum_{k'=1}^{K} w_{k,k'} W_1(P_S^k, P_T^{k'})$$

Thus we have,

$$\sum_{k=1}^{K} w_T^k W_1(P_S^k, P_T^k) \leq \min_{w \in \Pi(\mathbf{w_S}, \mathbf{w_T})} \sum_{k=1}^{K} \sum_{k'=1}^{K} w_{k,k'} W_1(P_S^k, P_T^{k'}) \tag{34}$$

$$= MW_1(P_S, P_T)$$

Also we prove in Theorem B.10 that:

$$MW_1(P_S, P_T) \leq W_1(P_S, P_T) + 4\sqrt{\epsilon}$$

Then we conclude our proof and show that:

$$\sum_{k=1}^{K} w_T^k W_1(P_S^k, P_T^k) \leq MW_1(P_S, P_T) \leq W_1(P_S, P_T) + 4\sqrt{\epsilon} \tag{35}$$

□

**Theorem B.12.** *Let* $\mathbb{H} \doteq \{f | f : \mathcal{X} \to [0,1]\}$ *denote a hypothesis space. Under the Assumptions in Theorem 4.7(or Theorem B.11), if the following assumption hold for all* $f \in \mathbb{H}$*:*

$$\sum_{k=1}^{K} w_T^k \gamma_S^k(f) \leq \sum_{k=1}^{K} w_S^k \gamma_S^k(f), \tag{36}$$

*then we have*

$$\sum_{k=1}^{K} w_T^k (\gamma^k)^\star \leq \gamma^\star.$$

*Further, let*

$$\epsilon_c(f) \doteq \sum_{k=1}^{K} w_T^k \gamma_S^k(f) + \sum_{k=1}^{K} w_T^k W_1(P_S^k, P_T^k) + \sum_{k=1}^{K} w_T^k (\gamma^k)^\star$$

*denote the sub-domain-based generalization bound, let*

$$\epsilon_g(f) \doteq \gamma_S(f) + W_1(P_S, P_T) + \gamma^\star$$

*denote the general generalization bound without any sub-domain information. We have for all* $f$*,*

$$\epsilon_c(f) \leq \epsilon_g(f) + \delta_c$$

*where* $\delta_c$ *is* $4\sqrt{\epsilon}$*.*

*Proof.* We will proove that $\sum_{k=1}^{K} w_T^k (\gamma^k)^\star \leq \gamma^\star$, where $\gamma^\star = \min_{f \in \mathbb{H}} \gamma_S(f) + \gamma_T(f)$, $\mathbb{H}$, and $(\gamma^k)^\star = \min_{f \in \mathbb{H}} \gamma_S^k(f) + \gamma_T^k(f)$

We have:

$$
\begin{aligned}
\gamma^\star &= \min_{f \in \mathbb{H}} \left( \gamma_S(f) + \gamma_T(f) \right) \\
&= \min_{f \in \mathbb{H}} \left( \sum_{k=1}^{K} w_S^k \gamma_S^k(f) + \sum_{k=1}^{K} w_T^k \gamma_T^k(f) \right) \\
&= \min_{f \in \mathbb{H}} \left( \sum_{k=1}^{K} w_S^k \gamma_{P_S^k}(f, g_S) + \sum_{k=1}^{K} w_T^k \gamma_{P_T^k}(f, g_T) \right) \\
&= \min_{f \in \mathbb{H}} \left( \sum_{k=1}^{K} w_T^k \gamma_{P_S^k}(f, g_S) + \sum_{k=1}^{K} w_T^k \gamma_{P_T^k}(f, g_T) + \sum_{k=1}^{K} w_S^k \gamma_{P_S^k}(f, g_S) - \sum_{k=1}^{K} w_T^k \gamma_{P_S^k}(f, g_S) \right) \\
&= \min_{f \in \mathbb{H}} \left( \sum_{k=1}^{K} w_T^k (\gamma_{P_S^k}(f, g_S) + \gamma_{P_T^k}(f, g_T)) + \sum_{k=1}^{K} (w_S^k - w_T^k) \gamma_{P_S^k}(f, g_S) \right) \\
&\geq \min_{f \in \mathbb{H}} \left( \sum_{k=1}^{K} w_T^k (\gamma_{P_S^k}(f, g_S) + \gamma_{P_T^k}(f, g_T)) \right) \\
&\geq \sum_{k=1}^{K} \min_{f \in \mathbb{H}} \left( \gamma_S^k(f) + \gamma_T^k(f) \right) \\
&= \sum_{k=1}^{K} w_T^k (\gamma^k)^\star
\end{aligned}
$$

$$\tag{37}$$

where the first inequality (the 6th line in the equation) is based on the assumption that $\sum_{k=1}^{K} w_T^k \gamma_S^k(f) \leq \sum_{k=1}^{K} w_S^k \gamma_S^k(f)$. The second inequality (the 7th line in the equation) is based on $\min\{f(x) + g(x)\} \geq \min\{f(x)\} + \min\{g(x)\}$

$\square$

## C   EMPIRICAL EVIDENCE FOR ASSUMPTIONS IN THEOREM 4.7

### C.1   GAUSSIAN DISTRIBUTION

To affirm the Gaussian distribution assumption for latent representations of each sub-domain, we include Figure 4. These plots, derived from the MNIST and MNIST-M datasets for all values of k, reinforce our theorem's practical realization.

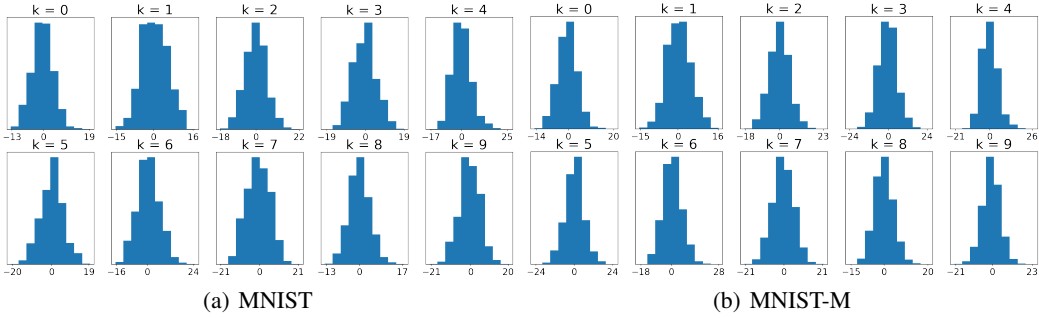

(a) MNIST  (b) MNIST-M

Figure 4: Distributions of the learned representations
.

## C.2 DISTANCE ASSUMPTION

We further present empirical evidence to support the distance relations between paired and non-paired source-target sub-domains on the MNIST to MNIST-M task. Our methodology specifically aims to minimize the distance between paired source-target sub-domains during the training process. Table 4 illustrates that empirically the distance between the paired source-target sub-domain is consistently less than the average distance between non-paired source-target sub-domains.

Table 4: Distance relations between paired and non-paired source-target sub-domains on the MNIST to MNIST-M task. We use Wasserstein-1 (W1) distance as distance metric.

| Sub-domain | Paired source-target sub-domain | Non-paired source-target sub-domains |
|---|---|---|
| 0 | 5.9 | 79.4 |
| 1 | 9.5 | 85.5 |
| 2 | 7.5 | 85.4 |
| 3 | 11.4 | 82.6 |
| 4 | 7.7 | 80.5 |
| 5 | 11.4 | 84.8 |
| 6 | 7.0 | 89.5 |
| 7 | 11.9 | 89.6 |
| 8 | 8.1 | 72.8 |
| 9 | 9.6 | 81.2 |

## D ALGORITHM

Our framework is outlined in pseudo-code in Algorithm 1.

---

**Algorithm 1** Domain Adaptation via Rebalanced Sub-domain Alignment(DARSA)

---

**Input:** Source data $X_S$; Source label $y_S$, Target data $X_T$; coefficient $\lambda_Y, \lambda_D, \lambda_c, \lambda_a$; learning rate $\alpha$;

Pretrain feature extractor and classifier with $X_S$ and $y_S$, initialize $\theta_E^S, \theta_E^T$, and $\theta_Y$ with pretrained weights. Initialize $w_T^k$ and $w_T^k$ with 1/K for k = 1,2 ..., K

**repeat**

    Sample minibatch from $X_S$ and $X_T$

    $\theta_Y \leftarrow \theta_Y - \alpha \nabla_{\theta_Y}(\lambda_Y \mathcal{L}_Y + \lambda_D \mathcal{L}_D + \lambda_c \mathcal{L}_{intra} + \lambda_a \mathcal{L}_{inter})$

    $\theta_E^S \leftarrow \theta_E^S - \alpha \nabla_{\theta_E^S}(\lambda_Y \mathcal{L}_Y + \lambda_D \mathcal{L}_D + \lambda_c \mathcal{L}_{intra} + \lambda_a \mathcal{L}_{inter})$

    $\theta_E^T \leftarrow \theta_E^T - \alpha \nabla_{\theta_E^T}(\lambda_D \mathcal{L}_D + \lambda_c \mathcal{L}_{intra} + \lambda_a \mathcal{L}_{inter})$

**until** $\theta_E^S, \theta_E^T$, and $\theta_Y$ converge

---

## E ANALYSIS OF FEATURE SPACE

We visualize the feature spaces learned by DANN (Ganin et al., 2016), CAT (Deng et al., 2019) and our method, DARSA, using UMAP (Sainburg et al., 2021). As shown in Figure 5, features learned

with DARSA form stronger clusters when the labels are the same, and clusters with different labels are more separated from one another. In contrast, both DANN and CAT fail to learn a good source-target domain alignment in the feature space (shown in Figure 5 (b)(c)) in the presence of label distribution shifts. This confirms that our method, DARSA, can learn a label-conditional feature space that is discriminative and domain-invariant, which improves performance in target domain prediction.

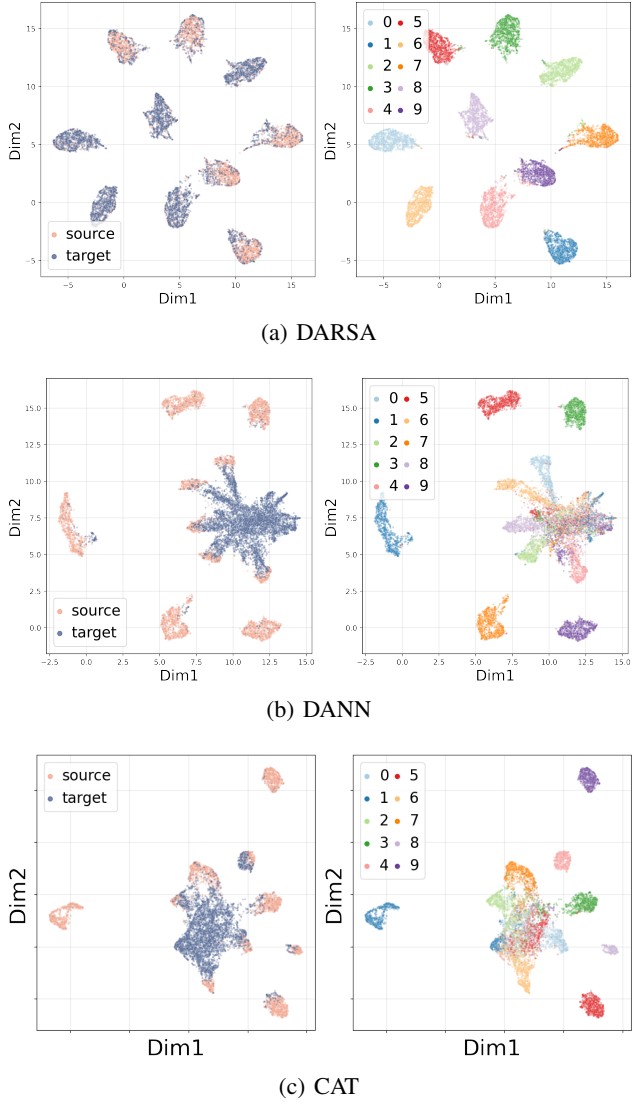

Figure 5: For MNIST to MNIST-M UDA task with label shifting (a) feature space learned by our method, DARSA. (b) feature space learned by DANN. (c) feature space learned by CAT. Left panel: colored by source/target; Right panel: colored by true label (digits). The features are projected to 2-D using UMAP.

## F  ANALYSIS OF WEIGHTS

In order to assess the accuracy of our target weight estimation in the DARSA algorithm, we conducted an additional analysis. This analysis focused on the MNIST to MNIST-M experiment under

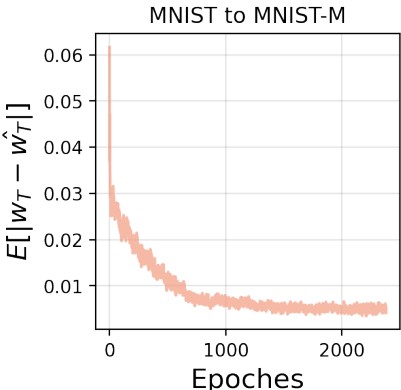

Figure 6: Evolution of the absolute difference between the estimated target weight $\hat{w_T}$ and the actual ground truth $w_T$ across epochs.

conditions of weak imbalance. Specifically, we compared the difference between the actual ground truth target weight (denoted as $w_T$) and our estimated target weight (denoted as $\hat{w_T}$) across epochs.

As illustrated in the figure 6, our estimation aligns closely with the ground truth towards the end of the training process. This proximity indicates the effectiveness of our weight estimation approach within the DARSA algorithm.

## G  DETAILS OF EXPERIMENTAL SETUP: DIGITS DATASETS WITH LABEL SHIFTING

In our Digits datasets experiments, we evaluate our performance across four datasets: MNIST (Le-Cun et al., 1998), MNIST-M (Ganin et al., 2016), USPS, and SVHN, all modified to induce label distribution shifts. Here, the parameter $\alpha$ denotes the class imbalance rate, representing a ratio such as 1:$\alpha$ and $\alpha$:1 for the odd:even distribution in the source and target datasets, respectively. Weak and strong imbalance correspond to $\alpha = 3$ and $\alpha = 8$. A small subset of the target domain is used for hyperparameter search, serving as an upper performance bound for UDA methods. Results shown in Table 2 demonstrates DARSA's competitiveness in handling label shifting. It is worth noting that many state-of-the-art comparisons are not specifically tailored for shifted label distribution scenarios, potentially affecting their performance. To ensure a fair comparison, we use the Ax platform (Bakshy et al.; Letham et al., 2019) for automated hyperparameter tuning to maximize domain-shifting performance (More details in G.8). Additionally, DARSA performs well with varying imbalance rates (Table 5) and competes favorably in scenarios without label distribution shifts (Table 7).

### G.1  DETAILS OF THE DIGITS DATASETS WITH LABEL DISTRIBUTION SHIFTS

#### G.1.1  WEAK IMBALANCE: $\alpha = 3$

**MNIST → MNIST-M**: For source dataset, we randomly sample 36000 images from MNIST training set with odd digits three times the even digits. For target dataset, we randomly sample 6000 images from MNIST-M constructed from MNIST testing set, with even digits three times the odd digits. To create MNIST-M dataset, we follow the procedure outlined in Ganin et al. (2016) to blend digits from the MNIST over patches randomly extracted from color photos in the BSDS500 dataset (Arbelaez et al., 2010).

**MNIST-M → MNIST**: For source dataset, we randomly sample 36000 images from MNIST-M constructed from MNIST training set, with even digits three times the odd digits. For target dataset, we randomly sample 5800 images from MNIST testing set, with odd digits three times the even digits.

**USPS → MNIST**: For source dataset, we randomly sample 3600 images from USPS training set, with even digits three times the odd digits. For target dataset, we randomly sample 5800 images from MNIST testing set, with odd digits three times the even digits.

**SVHN → MNIST**: For source dataset, we randomly sample 30000 images from SVHN training set, with even digits three times the odd digits. For target dataset, we randomly sample 5800 images from MNIST testing set, with odd digits three times the even digits.

### G.1.2   STRONG IMBALANCE: $\alpha = 8$

**MNIST → MNIST-M**: For source dataset, we randomly sample 27000 images from MNIST training set with odd digits eight times the even digits. For target dataset, we randomly sample 4500 images from MNIST-M constructed from MNIST testing set, with even digits eight times the odd digits. To create MNIST-M dataset, we follow the procedure outlined in Ganin et al. (2016) to blend digits from the MNIST over patches randomly extracted from color photos in the BSDS500 dataset (Arbelaez et al., 2010).

**MNIST-M → MNIST**: For source dataset, we randomly sample 13500 images from MNIST-M constructed from MNIST training set, with even digits eight times the odd digits. For target dataset, we randomly sample 13500 images from MNIST testing set, with odd digits eight times the even digits.

**USPS → MNIST**: For source dataset, we randomly sample 2700 images from USPS training set, with even digits eight times the odd digits. For target dataset, we randomly sample 13500 images from MNIST testing set, with odd digits eight times the even digits.

**SVHN → MNIST**: For source dataset, we randomly sample 27000 images from SVHN training set, with even digits eight times the odd digits. For target dataset, we randomly sample 18000 images from MNIST testing set, with odd digits eight times the even digits.

### G.2   ADDITIONAL EMPIRICAL ANALYSIS OF OUR PROPOSED GENERALIZATION BOUND

Here we empirically evaluate the proposed generalization bound with weak imbalance ($\alpha = 3$) and strong imbalance ($\alpha = 8$). As shown in Figure 7 and Figure 8, our empirical results demonstrate that the sub-domain-based generalization bound in Theorem 4.5 is empirically much stronger than the non-sub-domain-based bound in Theorem 4.1.

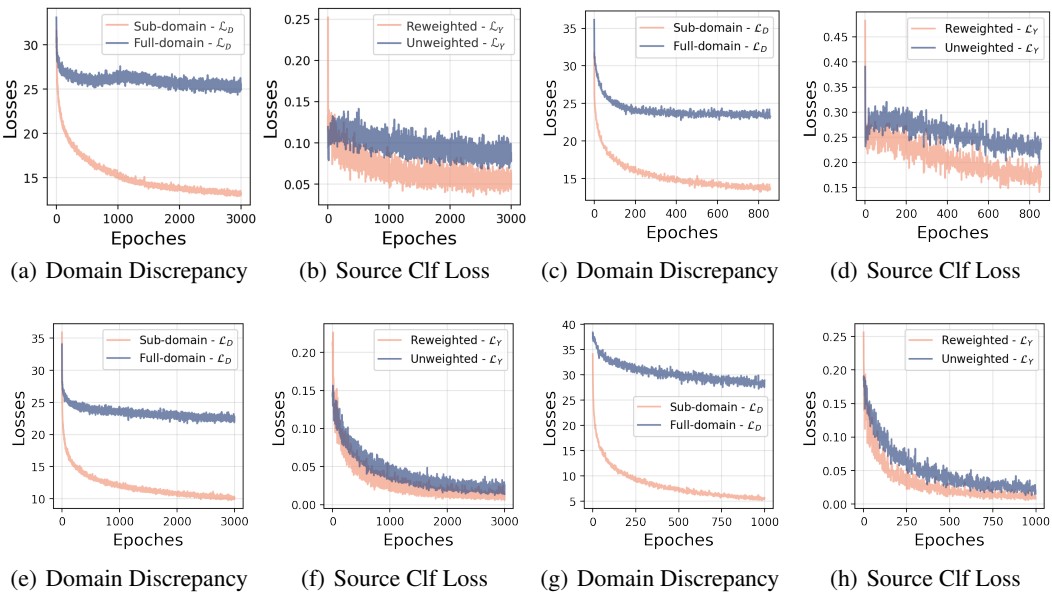

Figure 7: For experiments with weak imbalance ($\alpha = 3$), we compare the domain discrepancy term ($\mathcal{L}_D$) and source classification term ($\mathcal{L}_Y$) in our proposed bound to that in Theorem 4.1, respectively. Empirical results for each experiment are demonstrated in the subfigures: (a)(b) MNIST to MNIST-M (c)(d) MNIST-M to MNIST, (e)(f) USPS to MNIST, (g)(h) SVHN to MNIST.

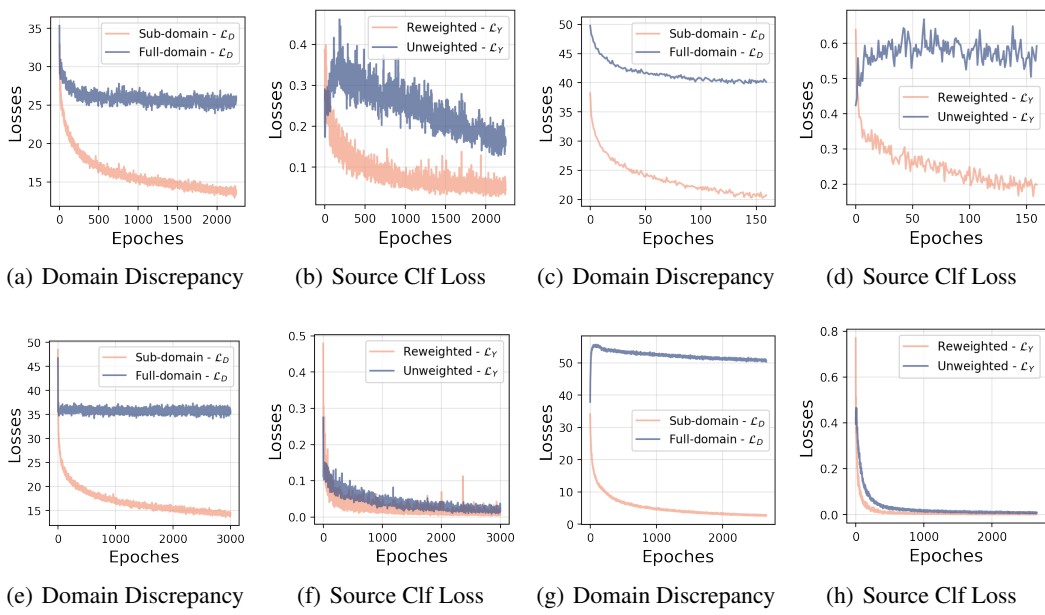

Figure 8: For tasks with strong imbalance ($\alpha = 8$), we compare the domain discrepancy term ($\mathcal{L}_D$) and source classification term ($\mathcal{L}_Y$) in our proposed bound to that in Theorem 4.1, respectively. Empirical results for each experiment are demonstrated in the subfigures: (a)(b) MNIST to MNIST-M (c)(d) MNIST-M to MNIST, (e)(f) USPS to MNIST, (g)(h) SVHN to MNIST.

### G.3 EVALUATE DARSA ON VARYING IMBALANCE RATES

We have conducted a study on the USPS to MNIST adaptation to explore the effects of varying imbalance rates. As can be seen from Table 5, the performance of our algorithm is stable across a wide range of imbalance rates.

Source: USPS - even : odd digit = 1:$\alpha$
Target: MNIST - odd : even digit = $\alpha$:1

Table 5: Summary of UDA results on USPS to MNIST adaptation with varying imbalance rates

| $\alpha$ | 1 | 2 | 3 | 4 | 5 | 6 | 7 | 8 | 9 | 10 |
|---|---|---|---|---|---|---|---|---|---|---|
| Accuracy (%) | 97.1 | 92.8 | 92.6 | 92.1 | 85.8 | 93.3 | 88.1 | 87.9 | 77.4 | 71.3 |

### G.4 THE RATIONALE FOR COMPETING ALGORITHMS CHOICES

For benchmark methods, we have chosen methods that not only have theoretical underpinnings, but also exhibit impressive performance in transfer learning on digit benchmarks as per the listings on the public competition Papers with Code (2023). Furthermore, our selection includes methods that contemplate sub-domain alignment to facilitate a direct comparison with our approach. It's important to note that our selection of benchmark methods may include some older models. However, these models were chosen not just for their performance but for their theoretical relevance and their ability to provide valuable insights into the effectiveness of our proposed method. Thus, while our experiments primarily serve to validate our theory, the benchmarks also offer meaningful evaluation of our theory's practical impact.

### G.5 MODEL STRUCTURES

For feature extractor, we employ a network structure similar to LeNet-5 ((LeCun et al., 1998)), but with minor modifications: the first convolutional layer produces 10 feature maps, the second convolutional layer produces 20 feature maps, and we use ReLU as an activation function for the hidden layer. Our feature space has 128 dimensions. For benchmarks, we utilize the network structures provided in the benchmark source code. In cases where experiments are not included in the source code, we use the same network architecture as our model to ensure fair comparisons. For classifier, we use a network structure with three fully connected layers with ReLU activation and a dropout layer with a rate of 0.5. See the included code link for further details of each experiment.

### G.6 ABLATION STUDY

To gain insight into the individual impact of each component within our objective function (Section 5), we performed an ablation study under conditions of weak imbalance. The outcomes of this investigation are detailed in Table 6. The ablation analysis confirms that each component in our objective function contributes to its overall performance. Therefore, we recommend the use of all components for optimal results. In addition, we have included ablation study with feature space visualization in Figure 9. As can be seen in Figure 9, the learned representation of DARSA has improved separation when using all the components, supporting the effectiveness of the proposed objective function.

### G.7 EVALUATE DARSA ON BENCHMARKS WITHOUT ADDITIONAL LABEL DISTRIBUTION SHIFTS

We conduct additional experiments to evaluate how DARSA performs in scenarios without label distribution shifts. Results in Table 7 show DARSA's performance is comparable with the most competitive methods.

### G.8 MODEL HYPERPARAMETERS

We use Adaptive Experimentation (Ax) platform (Bakshy et al.; Letham et al., 2019), an automatic tuning approaches to select hyperparameters to maximize the performance of our method. We use Bayesian optimization supported by Ax with 20 iterations to decide the hyperparameter choice. We note that most of the SOTA comparisons are not specifically designed for shifted label distribution scenarios, and this setting caused issues in several competing methods. We used Ax to maximize their performance in label shifting scenarios. Details on the model hyperparameters used for the Digits datasets with shifted label distribution are provided in Table 8 and Table 9 (If not explicitly

Table 6: Ablation study results. Each row represents a configuration with different $\lambda$ values. The last column reports the prediction accuracy (%) for each configuration.

| Experiment | $\lambda_Y$ | $\lambda_D$ | $\lambda_a$ | $\lambda_c$ | Accuracy |
|---|---|---|---|---|---|
| M → MM | 0.4 | 0.35 | 0.9 | 1 | 96.0 |
| M → MM | 0 | 0.35 | 0.9 | 1 | 61.3 |
| M → MM | 0.4 | 0 | 0.9 | 1 | 72.5 |
| M → MM | 0.4 | 0.35 | 0 | 1 | 61.9 |
| M → MM | 0.4 | 0.35 | 0.9 | 0 | 33.5 |
| MM → M | 1 | 0.5 | 1 | 1 | 98.8 |
| MM → M | 0 | 0.5 | 1 | 1 | 96.7 |
| MM → M | 1 | 0 | 1 | 1 | 98.4 |
| MM → M | 1 | 0.5 | 1 | 0 | 15 |
| MM → M | 1 | 0.5 | 0 | 1 | 98.2 |
| U → M | 1 | 0.5 | 1 | 1 | 92.60 |
| U → M | 0 | 0.5 | 1 | 1 | 65.90 |
| U → M | 1 | 0 | 1 | 1 | 85.80 |
| U → M | 1 | 0.5 | 0 | 1 | 76.20 |
| U → M | 1 | 0.5 | 1 | 0 | 58.40 |
| S → M | 0.95 | 0.11 | 0.11 | 0.3 | 90.1 |
| S → M | 0 | 0.11 | 0.11 | 0.3 | 77.9 |
| S → M | 0.95 | 0 | 0.11 | 0.3 | 86.1 |
| S → M | 0.95 | 0.11 | 0.11 | 0 | 64.3 |
| S → M | 0.95 | 0.11 | 0 | 0.3 | 84.9 |

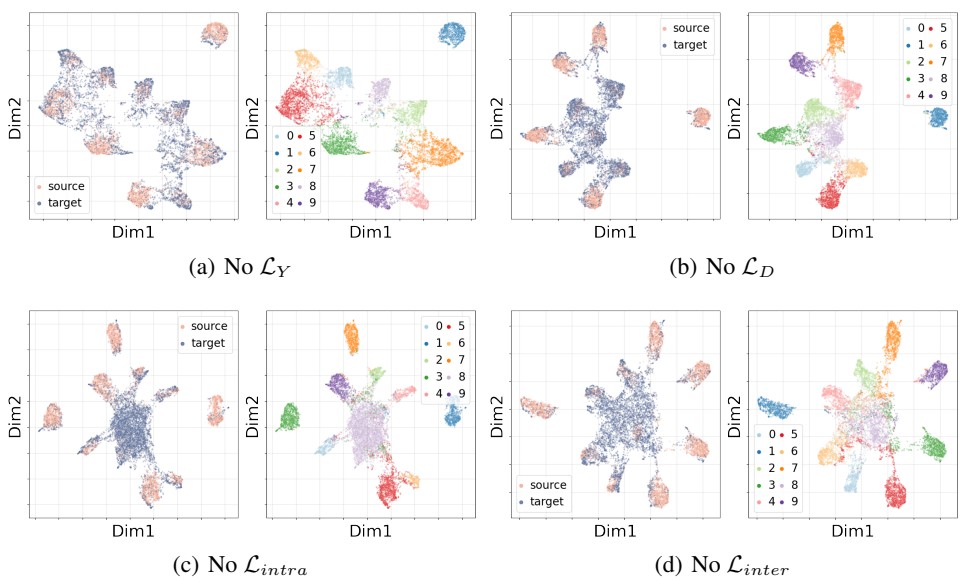

(a) No $\mathcal{L}_Y$        (b) No $\mathcal{L}_D$

(c) No $\mathcal{L}_{intra}$        (d) No $\mathcal{L}_{inter}$

Figure 9: For MNIST to MNIST-M UDA task with shifted label distribution with representations learnt by ablation study. Colored by 1) source/target; 2) predicted label (digit). The features are projected to 2-D using UMAP.

.

Table 7: Summary of UDA results on the Digits datasets without shifted label distribution, measured in terms of prediction accuracy (%) on the target domain.

| | DANN | WDGRL | DSN | ADDA | CAT | CDAN | pixelDA | DRANet | DARSA |
|---|---|---|---|---|---|---|---|---|---|
| U → M | 74.5 | 84.8 | 91.0 | 90.1 | 80.9 | 98.0 | 87.6 | 97.8 | 97.4 |
| S → M | 73.9 | 59.3 | 82.7 | 76.0 | 98.1 | 89.2 | 71.6 | 59.7 | 98.6 |

Table 8: Model hyperparameters used for Digits datasets with weak imbalance $\alpha = 3$

| | MNIST to MNIST-M | MNIST-M to MNIST | USPS to MNIST | SVHN to MNIST |
|---|---|---|---|---|
| DARSA | batch size = 512, $\alpha = 0.01, \lambda_Y = 0.4$, $\lambda_D = 0.35, \lambda_c = 1$, $\lambda_a = 0.9$, m = 30 SGD, momentum = 0.5 | batch size = 512, $\alpha = 0.01, \lambda_Y = 1$, $\lambda_D = 0.5, \lambda_c = 1$, $\lambda_a = 1$, m = 30 SGD, momentum = 0.4 | batch size = 256, , $\alpha = 0.01, \lambda_Y = 1$, $\lambda_D = 0.5, \lambda_c = 1$, $\lambda_a = 1$, m = 30 SGD, momentum = 0.4 | batch size = 256, $\alpha = 0.05, \lambda_Y = 0.95$, $\lambda_D = 0.11, \lambda_c = 0.3$, $\lambda_a = 0.11$, m = 50 SGD, momentum = 0.4 |
| DANN | batch size = 32 Adam, learning rate = 1e-4 | batch size = 32 Adam, learning rate = 1e-5 | batch size = 32 Adam, learning rate = 1e-4 | batch size = 64 Adam, learning rate = 1e-4 |
| WDGRL | batch size = 32 Adam, learning rate = 1e-5, $\gamma = 10$, critic training step: 1, feature extractor and discriminator training step: 3 | batch size = 64 Adam, learning rate = 1e-4, $\gamma = 10$, critic training step: 5, feature extractor and discriminator training step: 10 | batch size = 32 Adam, learning rate = 1e-4, $\gamma = 10$, critic training step: 1, feature extractor and discriminator training step: 2 | batch size = 32 Adam, learning rate = 1e-5, $\gamma = 10$, critic training step: 1, feature extractor and discriminator training step: 3 |
| DSN | batch size = 32 SGD, momentum = 0.8, learning rate = 1e-2, $\alpha = 0.01$, $\beta = 0.075, \gamma = 0.25$ | batch size = 32 SGD, momentum = 0.8, learning rate = 0.01, $\alpha = 0.01$, $\beta = 0.075, \gamma = 0.25$ | batch size = 32 SGD, momentum = 0.8, learning rate = 0.01, $\alpha = 0.01$, $\beta = 0.075, \gamma = 0.4$ | batch size = 512 SGD, momentum = 0.5, learning rate = 1e-5, $\alpha = 0.046$, $\beta = 0.61, \gamma = 0.92$ |
| ADDA | batch size = 64 Adam, learning rate = 1e-3, critic training step: 1, target model training step: 10 | batch size =64 Adam, learning rate = 1e-5, critic training step: 1, target model training step: 1 | batch size = 64 Adam, learning rate = 1e-3, critic training step: 1, target model training step: 1 | batch size = 64 Adam, learning rate = 1e-3, critic training step: 3, target model training step: 2 |
| CAT | batch size = 512 SGD, learning rate = $\frac{0.01}{(1+10p)^{0.75}}$, momentum = 0.9, p = 0.9, m = 30 | batch size = 128 SGD, learning rate = $\frac{0.01}{(1+10p)^{0.75}}$, momentum = 0.9, p = 0.9, m = 30 | batch size = 128 SGD, learning rate = $\frac{0.01}{(1+10p)^{0.75}}$, momentum = 0.9, p = 0.9, m = 30 | batch size = 256 SGD, learning rate = $\frac{0.01}{(1+10p)^{0.75}}$, momentum = 0.9, p = 0.9, m = 30 |
| CDAN | batch size = 64 SGD, momentum = 0.9, learning rate =0.01 | batch size = 64 SGD, momentum = 0.9, learning rate =0.01 | batch size = 64 SGD, momentum = 0.9, learning rate =0.01 | batch size = 64 SGD, momentum = 0.9, learning rate =0.1 |
| pixelDA | batch size = 64 Adam learning rate =0.0002, dim of the noise input: 10 | batch size = 64 Adam learning rate =0.0002, dim of the noise input: 10 | batch size = 32 Adam, learning rate =0.0001, dim of the noise input: 20 | batch size = 32 Adam, learning rate =0.0001, dim of the noise input: 20 |
| DRANet | batch size = 32 Adam | batch size = 32 Adam | batch size = 32 Adam | batch size = 32 Adam |
| MCD | batch size = 128 learning rate = 0.01 | batch size = 128 learning rate = 0.01 | batch size = 128 learning rate = 0.01 | batch size = 128 learning rate = 0.01 |
| MDD | batch size = 128 learning rate = 0.01 | batch size = 128 learning rate = 0.01 | batch size = 128 learning rate = 0.01 | batch size = 128 learning rate = 0.01 |

stated, we resort to the default hyperparameters from the respective implementations of the benchmark.).

Table 9: Model hyperparameters used for Digits datasets with strong imbalance $\alpha = 8$

| | MNIST to MNIST-M | MNIST-M to MNIST | USPS to MNIST | SVHN to MNIST |
|---|---|---|---|---|
| DARSA | batch size = 1024, $\alpha = 0.01$, $\lambda_Y = 0.8$, $\lambda_D = 0.4$, $\lambda_c = 0.9$, $\lambda_a = 0.9$, m = 30 SGD, momentum = 0.5 | batch size = 1024, $\alpha = 0.01$, $\lambda_Y = 1$, $\lambda_D = 0.5$, $\lambda_c = 1$, $\lambda_a = 0.5$, m = 30 SGD, momentum = 0.4 | batch size = 1024, , $\alpha = 0.01$, $\lambda_Y = 1$, $\lambda_D = 0.1$, $\lambda_c = 1$, $\lambda_a = 1$, m = 30 SGD, momentum = 0.4 | batch size = 1024, $\alpha = 0.01$, $\lambda_Y = 0.95$, $\lambda_D = 0.11$, $\lambda_c = 1$, $\lambda_a = 1$, m = 50 SGD, momentum = 0.4 |
| DANN | batch size = 64 Adam, learning rate = 1e-3 | batch size = 256 Adam, learning rate = 1e-6 | batch size = 64 Adam, learning rate = 1e-4 | batch size = 128 Adam, learning rate = 1e-3 |
| WDGRL | batch size = 64 Adam, learning rate = 1e-4, $\gamma = 10$, critic training step: 5, feature extractor and discriminator training step: 10 | batch size = 128 Adam, learning rate = 1e-5, $\gamma = 10$, critic training step: 1, feature extractor and discriminator training step: 5 | batch size = 64 Adam, learning rate = 1e-4, $\gamma = 10$, critic training step: 1, feature extractor and discriminator training step: 4 | batch size = 64 Adam, learning rate = 1e-6, $\gamma = 10$, critic training step: 1, feature extractor and discriminator training step: 3 |
| DSN | batch size = 32 SGD, momentum = 0.8, learning rate = 1e-2, $\alpha = 0.01$, $\beta = 0.075$, $\gamma = 0.25$ | batch size = 32 SGD, momentum = 0.8, learning rate = 1e-2, $\alpha = 0.01$, $\beta = 0.075$, $\gamma = 0.25$ | batch size = 64 SGD, momentum = 0.8, learning rate = 1e-2, $\alpha = 0.01$, $\beta = 0.075$, $\gamma = 0.4$ | batch size = 256 SGD, momentum = 0.1, learning rate = 1e-4, $\alpha = 0.046$, $\beta = 0.075$, $\gamma = 0.25$ |
| ADDA | batch size = 128 Adam, learning rate = 1e-5, critic training step: 1, target model training step: 1 | batch size =256 Adam, learning rate = 1e-6, critic training step: 2, target model training step: 1 | batch size = 128 Adam, learning rate = 1e-5, critic training step: 1, target model training step: 1 | batch size = 128 Adam, learning rate = 1e-6, critic training step: 4, target model training step: 1 |
| CAT | batch size = 512 SGD, learning rate = $\frac{0.01}{(1+10p)^{0.75}}$, momentum = 0.9, p = 0.9, m = 30 | batch size = 128 SGD, learning rate = $\frac{0.01}{(1+10p)^{0.75}}$, momentum = 0.9, p = 0.9, m = 30 | batch size = 256 SGD, learning rate = $\frac{0.01}{(1+10p)^{0.75}}$, momentum = 0.9, p = 0.9, m = 30 | batch size = 128 SGD, learning rate = $\frac{0.01}{(1+10p)^{0.75}}$, momentum = 0.9, p = 0.9, m = 30 |
| CDAN | batch size = 64 SGD, momentum = 0.9, learning rate =1e-3 | batch size = 128 SGD, momentum = 0.5, learning rate =0.1 | batch size = 64 SGD, momentum = 0.9, learning rate =0.01 | batch size = 16 SGD, momentum = 0.1, learning rate =1e-4 |
| pixelDA | batch size = 64 Adam learning rate =0.0002, dim of the noise input: 10 | batch size = 64 Adam learning rate =0.0002, dim of the noise input: 20 | batch size = 32 Adam, learning rate =0.001, dim of the noise input: 20 | batch size = 32 Adam, learning rate =0.001, dim of the noise input: 20 |
| DRANet | batch size = 32 Adam | batch size = 32 Adam | batch size = 32 Adam | batch size = 32 Adam |
| MCD | batch size = 128 learning rate = 0.1 | batch size = 128 learning rate = 0.01 | batch size = 128 learning rate = 0.01 | batch size = 128 learning rate = 0.01 |
| MDD | batch size = 128 learning rate = 0.1 | batch size = 128 learning rate = 0.01 | batch size = 128 learning rate = 0.01 | batch size = 128 learning rate = 0.01 |

# H DETAILS OF EXPERIMENTAL SETUP: TST DATASET WITH SHIFTED LABEL DISTRIBUTION

The Tail Suspension Test (TST) dataset (Gallagher et al., 2017) consists of local field potentials (LFPs) recorded from the brains of 26 mice. These mice belong to two genetic backgrounds: a genetic model of bipolar disorder (Clock-$\Delta$19) and wildtype mice. Each mouse is subjected to 3 behavioral assays which are designed to vary stress: home cage (HC), open field (OF), and tail-suspension (TS). We conduct experiments on two domain adaptation tasks using these neural activity data: transferring from wildtype mice to the bipolar mouse model and vice versa. We aim to predict for each one second window which of the 3 conditions (HC, OF, or TS) the mouse is currently experiencing. To create label distribution shifts, we subsample the datasets so that we have 6000 Homecage observations, 3000 Open Field observations, and 6000 Tail Suspension observations in the bipolar genotype dataset and 3000 Homecage observations, 6000 OpenField observations, and 3000 Tail Suspension observations in the wildtype genotype dataset.

### H.1 Details of the TST Dataset with Shifted Label Distribution

The Tail Suspension Test (TST) dataset (Gallagher et al., 2017) consists of 26 mice recorded from two genetic backgrounds, Clock-$\Delta$19 and wildtype. Clock-$\Delta$19 is a genotype which has been proposed as a model of bipolar disorder while wildtype is considered as a typical or common genotype. Local field potentials (LFPs) are recorded from 11 brain regions and segmented into 1 second windows. For each window, power spectral density, coherence, and granger causality features are derived. Each mouse is placed through 3 behavioral contexts while collecting LFP recordings: home cage, open field, and tail-suspension. Mice spent 5 minutes in the home cage which is considered a baseline or low level of distress behavioral context. Mice spent 5 minutes in the open field context which is considered a moderate level of distress. Mice then spent 10 minutes in the tail suspension test which is a high distress context.

### H.2 Model Structures

For feature extractor of the wildtype to bipolar task we use a network structure consisting of: a fully connected layer that maps our data to a feature space of 256 dimensions, with a LeakyReLU activation function; a fully connected layer that maps the feature space to 128 dimensions, and a Softplus activation function. For the bipolar to wildtype task, we use a network structure that includes: a fully connected layer that maps our data to a feature space of 256 dimensions, with a ReLU activation function; a fully connected layer that maps the feature space to 128 dimensions, with another ReLU activation function. For the classifier, we use a network structure that includes: three fully connected layers with ReLU activation and a dropout layer with a rate of 0.5. For benchmarks, we use the same network structures as our model to ensure fair comparisons, with the exception of DSN which has two fully connected layers with ReLU activation. See the included code link for additional details on each experiment.

### H.3 Model hyperparameters

Again, we use Adaptive Experimentation (Ax) platform (Bakshy et al.; Letham et al., 2019), an automatic tuning approaches to select hyperparameters to maximize the performance of our method. We use Bayesian optimization supported by Ax with 20 iterations to decide the hyperparameter choice. We note that most of the SOTA comparisons are not specifically designed for shifted label distribution scenarios, and this setting caused issues in several competing. We used Ax to maximize their performance in label shifting scenarios. Details on the model hyperparameters are provided in Table 10 (If not specified, the default hyperparameters from their respective implementations are employed.).

## I Accessibility of the Datasets and Computing Resources

**Accessibility of the Datasets**   The MNIST, BSDS500, USPS, and SVHN datasets are publicly available with an open-access license. The Tail Suspension Test (TST) dataset (Gallagher et al., 2017) is available to download at `https://research.repository.duke.edu/concern/datasets/zc77sr31x?locale=en` for free under a Creative Commons BY-NC Attribution-NonCommercial 4.0 International license.

**Computing Resources**   The experiments are conducted on a computer cluster equipped with a NVIDIA GeForce RTX 2080 Ti that has a memory capacity of 11019MiB.

Table 10: Model hyperparameters used for the label distribution shifting TST datasets

| | Clock-$\Delta$19 to Wildtype | Wildtype to Clock-$\Delta$19 |
|---|---|---|
| DARSA | batch size = 128, $\alpha$=1e-4, $\lambda_Y = 1$, $\lambda_D = 0.4$, $\lambda_c = 0.1$, $\lambda_a = 0.9$, m = 50 SGD, momentum = 0.6 | batch size = 128, $\alpha = 0.001$, $\lambda_Y = 0.7$, $\lambda_D = 0.1$, $\lambda_c = 0.1$, $\lambda_a = 1$, m = 50 SGD, momentum = 0.3 |
| DANN | batch size = 32 Adam, learning rate = 1e-4 | batch size = 32 Adam, learning rate = 1e-4 |
| WDGRL | batch size = 32 Adam, learning rate = 1e-4, $\gamma = 10$, critic training step: 1, feature extractor and discriminator training step: 2 | batch size = 32 Adam, learning rate = 1e-5, $\gamma = 10$, critic training step: 1, feature extractor and discriminator training step: 3 |
| DSN | batch size = 64 SGD, momentum = 0.5, learning rate = 0.1, $\alpha = 1$, $\beta = 1, \gamma = 1$ | batch size = 32 SGD, momentum = 0.5, learning rate = 0.1, $\alpha = 1$, $\beta = 1, \gamma = 1$ |
| CAT | batch size = 128 SGD, learning rate = $\frac{0.01}{(1+10p)^{0.75}}$, momentum = 0.9, p = 0.9, m = 3 | batch size = 128 SGD, learning rate = $\frac{0.01}{(1+10p)^{0.75}}$, momentum = 0.9, p = 0.9, m = 3 |
| CDAN | batch size = 64 SGD, momentum = 0.9, learning rate = 0.1 | batch size = 64 SGD, momentum = 0.9, learning rate = 0.1 |

