# OpenReview forum: "Understanding and Robustifying Sub-domain Alignment for Domain Adaptation"
_ICLR.cc/2024/Conference — Submitted to ICLR 2024_

### Official Review · Reviewer_p9oY · 2023-10-28

**Soundness:** 3 good
**Presentation:** 3 good
**Contribution:** 3 good
**Rating:** 6
**Confidence:** 4

**Summary:**

This paper establishes a theoretical foundation for sub-domain alignment in domain adaptation. Under certain plausible assumptions, they demonstrate that this bound is as tight, if not tighter, than those of full-domain alignment approaches. Drawing from this theorem, they introduce the Domain Adaptation via Rebalanced Sub-domain Alignment (DARSA) to address the challenge of marginal sub-domain weight shifts. Empirical evidence shows that their method, with sub-domains based on class labels, outperforms other leading domain adaptation methods in label shift scenarios.

**Strengths:**

1. The theoretical analysis regarding the sub-domain alignment is thorough and insightful. The method proposed is clearly grounded in this theory. While I haven't delved into every detail of the derivation, the conclusions seem sound.
2. The paper is well-structured and reader-friendly.
3. The effectiveness of their approach, when segmenting by class to form subdomains, is confirmed by results on both Digits and TST for the label shift scenario.

**Weaknesses:**

1. While the theory behind sub-domain alignment is certainly compelling, in its practical application, it merely utilizes class labels to segment subdomains. This results in a method that essentially merges class importance weighting with W1 distance to gauge domain discrepancies. While this is a fresh approach, it's not exceptionally innovative. It would enhance the paper if the author explored alternative sub-domain segmentation techniques.

2. It would enrich the study if the author included baselines that are specifically designed for label shift scenarios, like the method outlined in [1]. Many of their referenced baselines, such as DANN and ADDA, are primarily constructed for the covariate shift setting.


[1] Yan, Hongliang, et al. "Mind the class weight bias: Weighted maximum mean discrepancy for unsupervised domain adaptation." Proceedings of the IEEE conference on computer vision and pattern recognition. 2017.

**Questions:**

See weaknesses.

---

> ### Author Response · Authors · 2023-11-18
>
> Dear Reviewer p9oY,
>
> Thank you for your comments. Please see below our response to your concerns.
>
> ---
> > While the theory behind sub-domain alignment is certainly compelling, in its practical application, it merely utilizes class labels to segment subdomains. This results in a method that essentially merges class importance weighting with W1 distance to gauge domain discrepancies. While this is a fresh approach, it's not exceptionally innovative. It would enhance the paper if the author explored alternative sub-domain segmentation techniques.
>
> We would like to clarify that the primary focus of this work was not on algorithmic novelty, but rather on addressing a critical challenge observed in imbalanced scenarios. Our analysis revealed that subdomain-based methods tend to falter under such conditions. To counter this, we introduced an algorithm with **theoretical guarantee** to enhance the robustness of sub-domain based methods. The effectiveness of our approach is not just theoretical but is also supported by empirical evidence from our study. With the theoretical foundation provided by our work, we expect following works with more advanced algorithms.
>
> > It would enrich the study if the author included baselines that are specifically designed for label shift scenarios, like the method outlined in [1]. Many of their referenced baselines, such as DANN and ADDA, are primarily constructed for the covariate shift setting.
>
> **Baselines.** We have indeed included baselines specifically designed for label shift scenarios, such as Cluster Alignment with a Teacher (CAT). Since we are a theory-focused paper to theoretically understand the effectiveness of sub-domain based methods, our selection also includes theoretically robust methods such as DANN as well as top-performing methods in digit transfer learning tasks, recognized on the Papers with Code website [2]. It's worth noting that most of our domain adaptation accuracy is strong (over 90%), which naturally limits the margin for further improvement by adding additional benchmarks.
> We address a different level of imbalance compared to [1]. In [1], the focus is on the original class distributions of digit datasets, highlighting biases across domains (for example, the higher weights of classes 0 and 1 in USPS, and 1 and 2 in SVHN). These scenarios can be considered as **no additional label distribution shifts**, which is significantly different from the experiment setup in our work. In our work, we modified digit datasets to induce label distribution shifts. Here, the parameter $\alpha$ denotes the class imbalance rate, representing a ratio such as 1:$\alpha$ and $\alpha$:1 for the odd:even distribution in the source and target datasets, respectively. And we evaluated both weak ($\alpha=3$) and strong imbalance scenarios (($\alpha=8$)).
> In line with your suggestion, we have demonstrated experiments with **no additional label distribution shifts** for a more direct comparison of DARSA with other methods. The results, presented in the subsequent tables, clearly demonstrate that DARSA significantly outperforms [1], reinforcing the effectiveness and robustness of our approach in these specific scenarios.
> |                       |     USPS to MNIST         |  SVHN to MNIST  |
> | ------------------:| :------------------------------:| :----------------------:|
> |  DANN     |   	  74.5 	                      |   73.9                 |
> | **WDAN**  [1]     |    65.4	                      |   67.4                 |
> |  WDGRL |      84.8                           |    59.3                  |
> |  DSN        |      91.0                           |    82.7                |
> |  ADDA       |      90.1                        |    76.0                |
> |  CAT          |      80.9                         |   98.1                 |
> |  CDAN   |     98.0                         |   89.2                 |
> |  pixelDA   |     87.6                           |   71.6                   |
> |  DRANet  |     97.8                        |   59.7                   |
> |  DARSA          |     97.4                          |    98.6                   |
>
> Table: Performance of DARSA and baseline methods without label shifting.
>
> ---
>
> [1] Yan, Hongliang, et al. "Mind the class weight bias: Weighted maximum mean discrepancy for unsupervised domain adaptation." Proceedings of the IEEE conference on computer vision and pattern recognition. 2017.
>
> [2] https://paperswithcode.com/task/domain-adaptation

---

> > ### Comment · Reviewer_p9oY · 2023-11-22
> > **Thanks for your comments**
> >
> > Thanks for your comments. I will keep my score as 6.

---

> > > ### Author Response · Authors · 2023-11-22
> > >
> > > Thank you once again for the invaluable time and effort you put into reviewing our work. We are sincerely grateful for your insightful comments!

---

> > > ### Author Response · Authors · 2023-11-23
> > > **Thank you**
> > >
> > > Dear Reviewer p9oY,
> > >
> > > As we approach the conclusion of this discussion period, we would like to express our gratitude towards your constructive comments and dedicated efforts to help improve the quality of work.
> > >
> > > At this critical moment for our research, your expertise and support are immensely valuable to us. With great respect, we earnestly request your favorable consideration in advancing our work. As commented by you and your fellow reviewers, this work is well-written and establishes fundamental theoretical support for sub-domain based. Moreover, we believe it is valuable to have an algorithm that has strong performance on many real-life applications (e.g., medical applications where the number of labels is moderate) along with a safety lock (theoretical guarantees).
> > >
> > > The dedication of reviewers like you is what contributes to ICLR's esteemed reputation. Regardless of the final decision, we sincerely appreciate your help along the rebuttal process!
> > >
> > > Best,
> > >
> > > The Authors of Paper 8560

---

### Official Review · Reviewer_epCK · 2023-10-31

**Soundness:** 2 fair
**Presentation:** 3 good
**Contribution:** 2 fair
**Rating:** 5
**Confidence:** 4

**Summary:**

This paper first presents a theoretical analysis to establish that the sub-domain based methods are in fact optimizing a generalization bound that is at least as strong as the full-domain-based objective functions. Besides, the paper presents a UDA algorithm, Domain Adaptation via Rebalanced Sub-domain Alignment (DARSA), which addresses the case when marginal subdomain weights shift. DARSA optimizes reweighted classification error and discrepancy between sub-domains of the source and target task.

**Strengths:**

(1) This work provides a theoretical foundation for subdomain-based methods in domain adaptation, addressing their previous lack of rigorous understanding.
(2) The authors design a DARSA model, aiming to address shifted marginal sub-domain weights, which adversely impact existing sub-domain based methods.
(3) The experimental results on different benchmarks show the effectiveness of the proposed framework over other existing UDA methods.

**Weaknesses:**

(1) Most of the compared UDA methods were published before 2020. To fully validate the superiority of the proposed DARSA model, more SOTA UDA methods should be included in comparison experiments.

(2) More insightful analyses should be provided. For example, the visualization of the subdomain rebalancing weights should be shown.

(3) The current experiments are conducted on two datasets, which is not sufficient. It is recommended to conduct on large-scale datasets.

**Questions:**

(1) Most of the compared UDA methods were published before 2020. To fully validate the superiority of the proposed DARSA model, more SOTA UDA methods should be included in comparison experiments.

(2) More insightful analyses should be provided. For example, the visualization of the subdomain rebalancing weights should be shown.

(3) The current experiments are conducted on two datasets, which is not sufficient. It is recommended to conduct on large-scale datasets.

---

> ### Author Response · Authors · 2023-11-18
>
> Dear Reviewer epCK,
>
> Thank you for your comments. Please see below our response to your concerns.
>
> ---
>
> ### *Responses to Weakness/Question (1) and (3)*
> **Rationale Behind Baseline Selection:**  Regarding the selection of compared methods, our benchmarks were meticulously chosen to demonstrate the robustness and effectiveness of our proposed algorithm. This includes theoretically solid methods such as DANN, and methods that have top performances in digit transfer learning tasks as recognized on the Papers with Code website  [1]. We also included relevant sub-domain-based methods [2, 3], aligning perfectly with our research's core focus. This strategic selection of baselines was not only comprehensive but also directly relevant to our algorithm's intended application area.
>
> **Rationale Behind Datasets Selection:** Regarding the number of datasets used, our focus was on depth and relevance rather than quantity. The datasets selected are highly pertinent to the real-world applications we aim to impact. Specifically, our theoretical analysis indicates that DARSA should perform competitively in scenarios with a manageable number of classes, which is particularly applicable to fields like medical applications. These areas demand high precision and reliability, qualities that our algorithm has been demonstrated to fulfill effectively.
>
> In addition, as a theory-focused paper, we conducted comprehensive examinations to validate our theoretical propositions. This includes 1) empirically confirming the superiority of the sub-domain-based generalization bound; 2) verifying that the assumptions for Theorem 4.10 are empirically satisfied in real-world datasets; 3) demonstrating the vital role of subdomain weight re-balancing and showing DARSA's robustness to minor weight estimation discrepancies (Section 6.2); and 4) conducting an ablation study to evaluate the impact of each component within our objective function.
>
> ---
>
> ### *Responses to Weakness/Question (2)*
>
> It appears that the reviewer has neglected the analysis provided in our submitted manuscript. Section 6.2 presents experiments on the Digits datasets to demonstrate the importance of (i) weights re-weighting and (ii) the accuracy of target subdomain weights estimation. As shown in Table 1, while the oracle case provides the best performance, DARSA algorithm demonstrates a very similar level of effectiveness, demonstrating that the weights in the DARSA algorithm provide nearly the same quality of predictions. In addition, it can be seen  from Table 1 that DARSA is robust to minor divergence in weights estimation and varying imbalance rates.
>
> In addition, we would like to clarify that the primary indicator of the effectiveness of our method, DARSA, is the improvement in prediction accuracy, which directly reflects the precision of our estimated weights. However, to directly address your request and provide further transparency, we have included an additional analysis. This analysis involves comparing the difference between the actual ground truth target weight $w_T$ and our estimated target weight $\hat{w_T}$ across epochs.
>
> As illustrated in the figure 6 in Appendix F, our estimation aligns closely with the ground truth towards the end of the training process. This proximity indicates the effectiveness of our weight estimation approach within the DARSA algorithm.
>
> ---
>
> [1] https://paperswithcode.com/task/domain-adaptation
>
> [2] Deng, Zhijie, Yucen Luo, and Jun Zhu. "Cluster alignment with a teacher for unsupervised domain adaptation." Proceedings of the IEEE/CVF international conference on computer vision. 2019.
>
> [3] Long, Mingsheng, et al. "Conditional adversarial domain adaptation." Advances in neural information processing systems 31 (2018).

---

> ### Comment · Reviewer_epCK · 2023-11-22
>
> Thanks for your comments. Some concerns have been addressed, and thus I raise my score.

---

> ### Author Response · Authors · 2023-11-22
>
> Dear Reviewer epCK,
>
> As the discussion period approaches its end, we would like to again express our sincere gratitude for your help and dedication to the review of our work. In light of our discussions, we wish to highlight the core contributions that we believe are important to the field of domain adaptation research:
>
> **Main Contributions:**
>
> a. Theoretical Contributions:
>
> Our work analyzes and provides a theoretical foundation for sub-domain based methods in domain adaptation, addressing their previous lack of rigorous understanding. **Our theoretical framework not only supports our algorithm but can be extended to other methods, contributing to broader impact and value in the field.**
>
> b. Algorithmic Strengths:
>
> Our theoretical analysis leads to our algorithm DARSA, showing superior performance in various real-world applications. We also detail the circumstances where our algorithm excels or may face challenges. **This transparency, along with potential strong performance in applications such as medical applications, makes our algorithm highly valuable.** Specifically, our theoretical analysis indicates that DARSA should perform competitively in scenarios with a manageable number of classes, which is particularly applicable to fields like biomedical applications. For example, as illustrated in Table 3, our experiments on a neural biology dataset involve transferring knowledge between a bipolar disorder model (Clock-∆19) and control mice, each subjected to three behavioral assays. This scenario exemplifies the typical biomedical context with a manageable number of classes. Similarly, in predicting diabetes types among patients from different hospitals, we generally deal with a limited set of categories (type 1, type 2, and gestational diabetes). These cases highlight the practical significance of our algorithm in fields that demand high precision and reliability.
>
> We believe that our paper's blend of theoretical rigorousness, novel methods, and convincing empirical results represents a significant contribution to the field of domain adaptation. We are delighted that our effort has addressed some of your concerns, and deeply grateful for your recognition of our rebuttal, as reflected in the raised scores.
>
> Best regards,
>
> The Authors of Paper 8560

---

### Official Review · Reviewer_W1Lh · 2023-11-02

**Soundness:** 3 good
**Presentation:** 3 good
**Contribution:** 2 fair
**Rating:** 6
**Confidence:** 3

**Summary:**

This paper focuses on the understanding and robustifying of the application of sub-domain alignment to unsupervised domain adaptation. To achieve this goal, this paper provides a theoretical foundation for sub-domain alignment methods and an algorithm DARSA to address the shifted marginal sub-domain weights. The experiments show that the proposed method outperforms some previous methods.

**Strengths:**

(1)This paper is well-written, and it is enjoyable to read.
(2)The proposed algorithm is simple yet effective to some extent.

**Weaknesses:**

(1)The theoretical analysis is not instructive. The theoretical analysis has two main theorems, Sub-domain-based Generalization Bound (Theorem 4.5) and Benefits of Sub-domain Alignment (Theorem 4.7 and 4.10). Theorem 4.5 shows that the error of the target domain can be bounded by the weighted error of the source domain and the weighted error of the sub-domain distance. However, some works [1][2] have pointed out similar conclusions. Theorem 4.10 shows that the error bound of Theorem 4.5 is at least as strong as the full domain generalization bound without the sub-domain information. However, the authors may overlook the finiteness of real datasets, which is also important for reliable generalization bound and thus may lead to a different conclusion. Considering that the sub-domains have fewer samples than the whole domain, the finiteness improves more value of the sub-domain generalization bound than the whole-domain generalization bound.
(2)Although the paper is based on previous sub-domain alignment works, the previous works are not well connected in theoretical analysis, algorithm, or experiments. The author should give some examples of how the work enhances the understanding and robustifying of previous methods.
(3)The experiments are not sufficient to show the effectiveness of the proposed algorithm. First, the number of datasets used for the experiments is too small. Second, the compared methods are not advanced enough.
[1] Jiang X, Lao Q, Matwin S, et al. Implicit class-conditioned domain alignment for unsupervised domain adaptation[C]//International Conference on Machine Learning. PMLR, 2020: 4816-4827.
[2] Wen J, Greiner R, Schuurmans D. Domain aggregation networks for multi-source domain adaptation[C]//International conference on machine learning. PMLR, 2020: 10214-10224.

**Questions:**

As the weakness above

**Details Of Ethics Concerns:**

I have no ethics concerns.

---

> ### Author Response · Authors · 2023-11-18
>
> Dear Reviewer W1Lh,
>
> Thank you for your comments. Please see below our response to your concerns.
> > The theoretical analysis is not instructive. The theoretical analysis has two main theorems, Sub-domain-based Generalization Bound (Theorem 4.5) and Benefits of Sub-domain Alignment (Theorem 4.7 and 4.10). Theorem 4.5 shows that the error of the target domain can be bounded by the weighted error of the source domain and the weighted error of the sub-domain distance. However, some works [1][2] have pointed out similar conclusions.
>
> We indeed have already included [1] in our Related Work section. This is one great example of sub-domain-based domain adaptation method that uses class-conditioned domain alignment to address within-domain class imbalance and between-domain class distribution shift. Our theoretical framework provides a foundation for subdomain based methods, addressing their previous lack of rigorous understanding. One of the key messages we hope to deliver is that **Our theoretical framework not only supports our algorithm but can be extended to other methods, contributing to broader impact and value in the field.**  Contrasting with the focus of [2] on **multi-source** domain adaptation, where source domains are aggregated based on their relevance to the target domain (with weights assigned accordingly), our work focuses on **single source** domain adaptation. Specifically, we focus on aligning corresponding sub-domain distributions. For example, we align label 1 in the source domain with label 1 in the target domain. Our weighting logic also differs – rather than relying on the relevancy of source sub-domains, our method gives priority to the important sub-domains within the target domain. This fundamental difference underscores the uniqueness and strength of our methodology in addressing domain adaptation challenges.
>
> > Theorem 4.10 shows that the error bound of Theorem 4.5 is at least as strong as the full domain generalization bound without the sub-domain information. However, the authors may overlook the finiteness of real datasets, which is also important for reliable generalization bound and thus may lead to a different conclusion. Considering that the sub-domains have fewer samples than the whole domain, the finiteness improves more value of the sub-domain generalization bound than the whole-domain generalization bound.
>
> While the variance increases with fewer samples, this does not imply that the realized error bound is necessarily larger than the expected one because it is equally likely that the realized error bound is smaller than the expected one. We have proved that in expectation, the sub-domain-based generalization bound presents a tighter estimation. Moreover, please note that the classical domain generalization bound have about $O(n^{-1/2})$ concentration while in the sub-domain-based generalization bound, the averaged error have about $O(m^{-1})$ variance reduction where $m$ is the number of class. This leads to $O((mn)^{-1/2})$ concentration, i.e., the variance is in fact reduced by a multiple of $m$ compared to the non-subdomain-based one. Intuitively, this is similar to the classical result in information theory where conditioning on the class random variable can only decrease entropy. In addition, as shown in Section 6.1, our empirical results demonstrate that the subdomain-based generalization bound is empirically much stronger than the non-sub-domain-based bound, corroborating our insights for the effectiveness of sub-domain based methods.
>
> > Although the paper is based on previous sub-domain alignment works, the previous works are not well connected in theoretical analysis, algorithm, or experiments. The author should give some examples of how the work enhances the understanding and robustifying of previous methods.
>
> It appears that there is some misunderstanding that our paper primarily build upon previous sub-domain alignment works. Instead, our paper introduces a fresh perspective by analyzing and providing a much-needed theoretical foundation for sub-domain based methods which are popular yet theoretically underexplored. To further clarify, our theoretical analysis does not simply follow the trajectory of existing literature; it carves a new path in understanding how subdomain based methods function and can be optimized. This contributes not only to a deeper comprehension of these methods but also offers tools that can be used to enhance their robustness and efficacy. By doing so, our work extends its impact beyond just supporting our algorithm; it enriches the field of domain adaptation with a solid theoretical base that can be used to re-examine, refine, and strengthen existing subdomain based methods. We believe that such a theoretical foundation is important to the field and merits recognition.

---

> ### Author Response · Authors · 2023-11-18
> **Continuation**
>
> > The experiments are not sufficient to show the effectiveness of the proposed algorithm. First, the number of datasets used for the experiments is too small. Second, the compared methods are not advanced enough.
>
> **Rationale Behind Baseline Selection:**  Regarding the selection of compared methods, our benchmarks were meticulously chosen to demonstrate the robustness and effectiveness of our proposed algorithm. This includes theoretically solid methods such as DANN and methods that have top performances in digit transfer learning tasks as recognized on the Papers with Code website [1]. We also included relevant sub-domain-based methods [2, 3], aligning perfectly with our research's core focus. This strategic selection of baselines was not only comprehensive but also directly relevant to our algorithm's intended application area.
>
> **Rationale Behind Datasets Selection:** Regarding the number of datasets used, our focus was on depth and relevance rather than mere quantity. The datasets selected are highly pertinent to the real-world applications we aim to impact. Specifically, our theoretical analysis indicates that DARSA should perform competitively in scenarios with a manageable number of classes, which is particularly applicable to fields like medical applications. These areas demand high precision and reliability, qualities that our algorithm has been demonstrated to fulfill effectively.
>
> In addition, as a theory-focused paper, we conducted comprehensive examinations to validate our theoretical propositions. This includes 1) empirically confirming the superiority of the sub-domain-based generalization bound; 2) verifying that the assumptions for Theorem 4.10 are empirically satisfied in real-world datasets; 3) demonstrating the vital role of subdomain weight re-balancing and showing DARSA's robustness to minor weight estimation discrepancies (Section 6.2); and 4) conducting an ablation study to evaluate the impact of each component within our objective function.
>
> ---
>
> [1] https://paperswithcode.com/task/domain-adaptation
>
> [2] Deng, Zhijie, Yucen Luo, and Jun Zhu. "Cluster alignment with a teacher for unsupervised domain adaptation." Proceedings of the IEEE/CVF international conference on computer vision. 2019.
>
> [3] Long, Mingsheng, et al. "Conditional adversarial domain adaptation." Advances in neural information processing systems 31 (2018).

---

> > ### Comment · Reviewer_W1Lh · 2023-11-23
> >
> > Thanks for your comments.Most of my concerns have been addressed, I have modified my scores.

---

> > > ### Author Response · Authors · 2023-11-23
> > > **Thank you**
> > >
> > > Dear Reviewer W1Lh,
> > >
> > > As the discussion period draws to a close, we wish to express our sincere gratitude for your acknowledgement of our rebuttal, which confirms that most of the concerns have been adequately addressed.
> > >
> > > In these crucial stages of our research, the value of your expertise and backing cannot be overstated. We respectfully and eagerly seek your support in promoting our work further. Your observations, along with those from other reviewers, have highlighted the clarity and fundamental theoretical contributions of our manuscript, particularly in sub-domain based approaches.  Moreover, we believe it is valuable to have an algorithm that has strong performance on many real-life applications (e.g., medical applications where the number of labels is moderate) along with a safety lock (theoretical guarantees).
> > >
> > > Your commitment as a reviewer plays a significant role in upholding the distinguished reputation of ICLR. We are deeply grateful for your assistance throughout the rebuttal process, regardless of the outcome.
> > >
> > > Warm regards,
> > >
> > > The Authors of Paper 8560

---

### Meta-Review · Area_Chair_Emmk · 2023-12-09

**Metareview:**

This is a borderline paper. On one hand, this paper makes some contributions to a theoretical study about sub-domain alignment for DA. On the other hand, the empirical studies are relatively weak to comprehensively demonstrate the effectiveness of the proposed method based on the theoretical findings. Though the authors explain the rationales behind dataset selection and baseline selection, overall, the experimental setup and results are not convincing enough to fully demonstrate the effectiveness of a general DA method.

**Justification For Why Not Higher Score:**

Experimental results are not comprehensive or convincing enough.

**Justification For Why Not Lower Score:**

N/A

---

### Decision · Program_Chairs · 2024-01-16

Reject